# Plasma membrane transbilayer asymmetry of PI(4,5)P$_2$ drives unconventional secretion of Fibroblast Growth Factor 2

Manpreet Kaur ©, Fabio Lolicato © ✉ & Walter Nickel © ✉

Unconventional secretion of Fibroblast Growth Factor 2 (FGF2) is mediated by direct translocation across the plasma membrane. This process is initiated by PI(4,5)P$_2$-dependent FGF2 oligomerization at the inner plasma membrane leaflet. PI(4,5)P$_2$ is a non-bilayer lipid that accumulates at sites of FGF2 oligomerization, imposing severe membrane stress that is relieved by the formation of a lipidic membrane pore. At the outer leaflet, FGF2 oligomers are captured and disassembled by the heparan sulfate proteoglycan Glypican-1 (GPC1), making available FGF2 to engage in ternary signaling complexes on cell surfaces. Using an in vitro reconstitutions system, this study provides direct evidence that transbilayer asymmetry of PI(4,5)P$_2$ promotes rapid kinetics of membrane pore formation. Likewise, FGF2 secretion from cells is inhibited when PI(4,5)P$_2$ plasma membrane transbilayer asymmetry is disrupted. We propose the asymmetric distribution of PI(4,5)P$_2$ to lower the energetic barrier for membrane pore formation, enabling rapid kinetics of FGF2 membrane translocation into the extracellular space.

Fibroblast Growth Factor 2 (FGF2) belongs to a subclass of secretory proteins that do not travel along the ER/Golgi-dependent secretory route to reach the extracellular space. Such processes have collectively been termed "unconventional protein secretion" (UPS)[1–3]. Several alternative secretory routes have been discovered with FGF2 being the prime example of a UPS type I cargo protein that is secreted by direct translocation across the plasma membrane[4,5]. This process is initiated by sequential interactions of FGF2 with molecular components associated with the inner plasma membrane leaflet with (i) the Na,K-ATPase, (ii) Tec kinase, and (iii) the phosphoinositide PI(4,5)P$_2$[6–11]. The Na,K-ATPase and Tec kinase have been proposed to represent auxiliary factors promoting efficient secretion by accumulating FGF2 at the inner plasma membrane leaflet. In addition, Tec kinase may play a role in enhancing FGF2 secretion in a genuine cancer context with high levels of PI(3,4,5)P$_3$, resulting in enhanced membrane recruitment of Tec kinase. By contrast, PI(4,5)P$_2$ has been shown to be an essential component of the core machinery of unconventional secretion of FGF2. The interaction with PI(4,5)P$_2$ causes FGF2 to oligomerize on the surface of the inner plasma membrane leaflet, resulting in the opening of a lipidic membrane pore with a toroidal architecture through which

FGF2 translocates into the extracellular space[12–14]. This process is driven by the cell surface heparan sulfate proteoglycan Glypican-1 (GPC1) that, by direct competition for the binding site of PI(4,5)P$_2$ in FGF2, captures and disassembles FGF2 oligomers at the outer plasma membrane leaflet[15,16], making FGF2 available for entering ternary complexes with heparan sulfate chains and high-affinity FGF receptors for autocrine and paracrine signaling[17]. A number of UPS type I cargo proteins have been identified[2–5,18,19], including HIV-Tat and Engrailed2 that are secreted in a PI(4,5)P$_2$ and/or heparan sulfate dependent manner[20–23].

The mechanism underlying unconventional secretion of FGF2 has been reconstituted with purified components using an inside-out topology system based on giant unilamellar vesicles (GUVs)[12]. The minimal machinery driving FGF2 membrane translocation could be identified with PI(4,5)P$_2$ and heparin (used as a surrogate of heparan sulfates) on opposing sides of the membrane. Membrane translocation could only be observed with FGF2 containing all *cis* elements required for unconventional secretion from cells, such as the PI(4,5)P$_2$ binding pocket and C95 required for oligomerization[6,14,16,24]. FGF2 membrane translocation into the lumen of GUVs could be observed at a time scale

Heidelberg University Biochemistry Center, Heidelberg, Germany. ✉e-mail: fabio.lolicato@bzh.uni-heidelberg.de; walter.nickel@bzh.uni-heidelberg.de

of minutes[9,12]. In addition to in vitro approaches, individual events of FGF2 membrane translocation across plasma membranes have been visualized in intact cells, using real-time TIRF microscopy with single molecule resolution[25]. This study revealed extremely fast kinetics of FGF2 membrane translocation in a cellular context with an average time interval for individual events of about 200 milliseconds. Thus, a substantial difference in the kinetics of FGF2 membrane translocation could be observed between in vitro experiments under defined conditions and FGF2 secretion from intact cells[3].

Several parameters may play fundamental roles in governing the thermodynamic principles underlying fast FGF2 membrane translocation kinetics observed in living cells[25]. Among them are the Na,K-ATPase as a component to accumulate FGF2 at the inner plasma membrane leaflet of intact cells, presumably in close spatial proximity to GPC1[26]. In addition, in cells, oxidative dimerization of FGF2 is likely to be mediated by an enzymatic mechanism, driving fast kinetics of PI(4,5)P$_2$-dependent oligomerization. A third potential factor is the asymmetric distribution of membrane lipids in plasma membranes. Sphingomyelin, for example, is predominantly found in the outer leaflet, whereas for example, phosphatidylethanolamine, phosphatidylinositol, and phosphatidylserine are enriched in the inner leaflet. This spatial organization is typically not preserved in artificial membranes such as GUVs used in reconstitution experiments of FGF2 membrane translocation[12]. In particular, the asymmetric distribution of PI(4,5)P$_2$ in native plasma membranes is a notable difference compared to its symmetric distribution in artificial GUVs and, therefore, may contribute to the vastly different membrane translocation kinetics of FGF2 when these systems were compared. Hence, in the current study, we tested the hypothesis that PI(4,5)P$_2$ transbilayer asymmetry reduces the energy costs of FGF2-dependent membrane pore formation.

The rationale for this was based on the concept of FGF2-dependent accumulation of the non-bilayer membrane lipid PI(4,5)P$_2$ to produce a steep and spatially constrained electrochemical gradient across the plasma membrane that may facilitate the transformation of the stable lipid bilayer into a highly transient lipidic membrane pore[27,28]. PI(4,5)P$_2$-dependent FGF2 recruitment and oligomerization in cholesterol-rich domains has been shown to increase membrane tension[13]. Based on our evidence-driven hypothesis that the FGF2 secretion machinery resides in liquid-ordered membrane domains[26], this increase in membrane tension represents another parameter that is likely to facilitate pore opening[29,30]. It is of note that a single FGF2 molecule is capable of recruiting up to 5 PI(4,5)P$_2$ molecules through one high and several additional low-affinity binding sites[12]. Since FGF2 is known to oligomerize in a membrane-dependent manner at least into tetramers[12,31–33], this would translate into the accumulation of a minimum of about 20 PI(4,5)P$_2$ molecules in a spatially constrained manner. This in turn would produce a massive amount of local membrane stress in the vicinity of membrane-bound FGF2 oligomers at the inner plasma membrane leaflet based on (i) a steep electrochemical gradient of PI(4,5)P$_2$ across the plasma membrane, (ii) a local accumulation of a lipid with non-bilayer characteristics and (iii) membrane tension based on high levels of cholesterol in regions of PI(4,5)P$_2$-dependent FGF2 recruitment. We therefore hypothesized that the local and asymmetric accumulation of PI(4,5)P$_2$ at FGF2 recruitment sites actively lowers the free energy barrier for the formation of a lipidic membrane pore in living cells.

To test this, we developed an experimental in vitro system to produce model membranes that contain an asymmetric transbilayer distribution of PI(4,5)P$_2$. Using an established assay to quantify FGF2-dependent membrane pore formation, we found an asymmetric distribution of PI(4,5)P$_2$ to accelerate this process. Consistently, in cell-based experiments, we demonstrated FGF2 secretion to be inhibited under conditions compromising PI(4,5)P$_2$ transbilayer asymmetry. Taken together, our findings provide direct evidence for PI(4,5)P$_2$

asymmetry to represent a critical determinant for efficient and fast FGF2 membrane translocation. Given the central role of PI(4,5)P$_2$ in regulating a wide array of cytosolic proteins, our results suggest that transbilayer asymmetry of this lipid may represent a general and tunable biophysical principle that lowers the energy barrier for membrane remodeling events.

## Results

### Formation of artificial membranes with an asymmetric transbilayer distribution of PI(4,5)P$_2$

To test the hypothesis of PI(4,5)P$_2$ transbilayer asymmetry to play a role in FGF2 membrane translocation, we aimed at producing artificial membranes characterized by PI(4,5)P$_2$ being present exclusively in the outer leaflet. We produced both Large Unilamellar Vesicles (LUVs) and Giant Unilamellar Vesicles (GUVs) containing PI(4)P, a phosphoinositide to which FGF2 does not bind[6,34]. As illustrated in Fig. 1A, the approach was to convert PI(4)P into PI(4,5)P$_2$ by an enzymatic reaction catalyzed by the lipid kinase PIP5K1C[35]. In the presence of Mg$^{2+}$ ions and ATP, assuming exclusion of the enzyme from the lumina of LUVs and GUVs, we aimed at the production of PI(4,5)P$_2$ exclusively in the outer leaflet of LUVs and GUVs. LUVs were used to establish the biochemical conditions for the enzymatic conversion of PI(4)P into PI(4,5)P$_2$. The procedure was then transferred to GUVs to run functional assays monitoring FGF2-dependent membrane pore formation with confocal microscopy as a read-out. For LUVs, two types of analyses were conducted to test successful production of PI(4,5)P$_2$ with (i) FGF2-GFP binding to LUVs analyzed by a previously established flow cytometry assay (Fig. 1B) and (ii) biochemical sedimentation assays (Fig. 1C). As demonstrated in Fig. 1B, in comparison to PI(4,5)P$_2$, FGF2-GFP does not bind to PI(4)P. Following the addition of Mg$^{2+}$ ions as part of the enzymatic conversion of PI(4)P into PI(4,5)P$_2$, relative binding efficiencies of FGF2-GFP were generally lower due to clustering of liposomes having an impact on quantification using flow cytometry. Nevertheless, following conversion of PI(4)P into PI(4,5)P$_2$, similar levels of FGF2-GFP binding were observed compared to LUVs to which PI(4,5)P$_2$ had been added during their formation. By contrast, under mock conditions, FGF2-GFP failed to bind to liposomes that were treated with ATP and Mg$^{2+}$ ions in the absence of PIP5K1C. Similar results were obtained when biochemical sedimentation experiments were done to quantify FGF2-GFP binding to LUVs (Fig.1C). With this read-out, FGF2-GFP binding efficiencies were similar under standard conditions and in the presence of ATP, and Mg$^{2+}$ ions and PIP5K1C. Upon enzymatic conversion of PI(4)P into PI(4,5)P$_2$, efficient FGF2-GFP binding could be observed that was similar to LUVs to which PI(4,5)P$_2$ had been added during their formation. Finally, we extended this system to GUVs analyzed by confocal microscopy (Fig. 1D, E). While FGF2-GFP did bind efficiently to PI(4,5)P$_2$-containing GUVs, it did interact with GUVs made with PI(4)P only after enzymatic conversion into PI(4,5)P$_2$. The combined experiments shown in Fig. 1 demonstrate the feasibility of the enzymatic conversion approach to generate PI(4,5)P$_2$ from PI(4)P in the outer leaflet of both LUVs and GUVs.

### Analysis of the transbilayer distribution of PI(4,5)P$_2$ in large unilamellar vesicles

To test whether liposomes formed by the experimental approach shown in Fig. 1 actually contain an asymmetric transbilayer distribution of PI(4,5)P$_2$, we established a quantitative procedure that was based on sequential disintegration and reformation of LUVs followed by FGF2-GFP binding experiments (Fig. 2). Large Unilamellar Vesicles (LUVs) were disintegrated using 40% methanol in chloroform. Lipids were extracted from the organic phase and were used to reform liposomes, followed by the quantitative assessment of FGF2-GFP binding to PI(4,5)P$_2$ using flow cytometry[6,34]. As illustrated in Fig. 2A, upon disintegration and reformation, we expected a reduction of surface PI(4,5)P$_2$ levels in asymmetric LUVs due to dilution of PI(4,5)P$_2$ into the

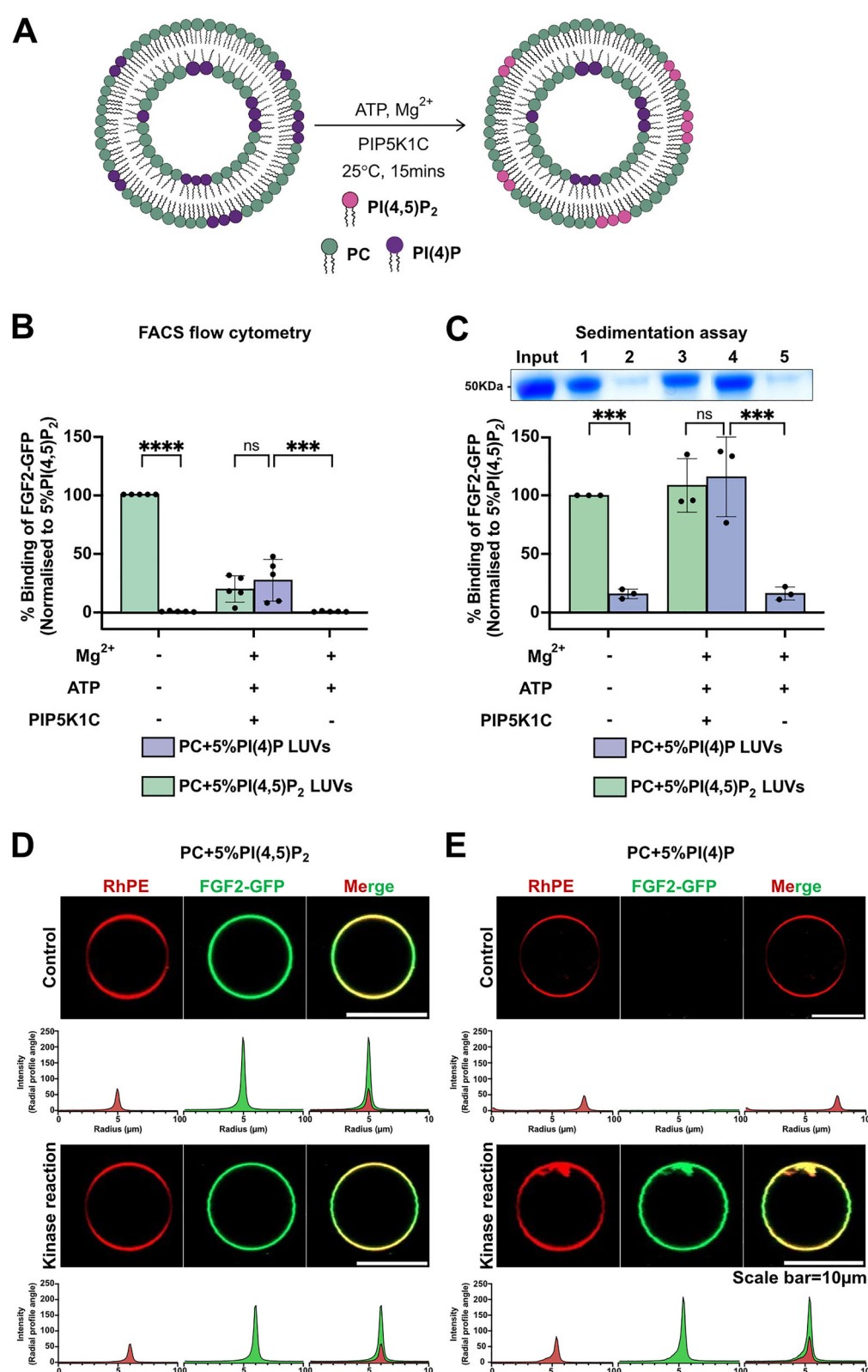

inner leaflet (Fig. 2A, lower panel). By contrast, a disintegration and reformation procedure would not alter the surface amounts of PI(4,5)P$_2$ in LUVs with a symmetric distribution of PI(4,5)P$_2$ (Fig. 2A, upper panel). As shown in Fig. 2B with non-treated LUVs shown in grey and disintegrated/reformed LUVs in light red, the binding efficiencies of FGF2-GFP remained unchanged for PI(4,5)P$_2$ and PI(4)P containing LUVs after reformation (Fig. 2B; conditions a versus b). LUVs

containing PI(4,5)P$_2$, treated with PIP5K1C, Mg$^{2+}$, and ATP, when measured without disintegration/reformation, showed lower binding efficiencies towards FGF2-GFP, a phenomenon that is due to Mg$^{2+}$-induced vesicle clustering (Fig. 2B, condition c, grey bar). However, following disintegration/reformation, the binding went up to normal levels (Fig. 2B, condition c, light red bar). This is because the disintegration/reformation procedure apparently removed Mg$^{2+}$ ions, as a result of

**Fig. 1 | Enzymatic approach to generate PI(4,5)P$_2$ asymmetric vesicles.**
**A** Schematic representation of enzymatic approach to generates transbilayer PI(4,5)P$_2$ asymmetry (adapted from ref. 56). Reaction utilizes PI(4)P containing liposomes with Mg$^{2+}$, ATP, and PIP5K1C to selectively phosphorylate PI(4)P to PI(4,5)P$_2$ on the outer leaflet of the vesicles. Large unilamellar vesicles (LUVs) were prepared with 5 mol% PI(4,5)P$_2$ or 5 mol% PI(4)P in PC background. LUVs with 5 mol% PI(4,5)P$_2$ were subjected to kinase reaction or left untreated. Similarly, LUVs with 5 mol% PI(4)P were subjected to kinase reaction or mock reaction (without PIP5K1C) or left untreated. After the reaction, vesicles were washed to remove unreacted components or damaged vesicles. They were incubated with FGF2-GFP for 60 min, and unbound protein was removed by centrifugation before analysis.
**B** Analysis of FGF2-GFP binding using FACS flow cytometry. Each dot represents a biological replicate; values are the mean of 2–3 technical replicates. For each experiment, average signal from 5 mol% PI(4,5)P$_2$ was set to 100%, and all values were normalized accordingly. Mean values with standard deviation (S.D.) for 5

biological replicates are presented. **C** Analysis of FGF2-GFP binding using Sedimentation assay. The bound fraction of FGF2-GFP was analyzed by SDS-PAGE and detected with Coomassie staining. Gel was imaged by EPSON 4870 scanner. Data were normalized to 5 mol% PI(4,5)P$_2$ within each experiment. Mean values with S.D. for 3 biological replicates are presented. **D** Giant unilamellar vesicles (GUVs) were prepared with 5 mol% PI(4,5)P$_2$ or **E** 5 mol% PI(4)P along with membrane marker Rhodamine-PE, in PC background. GUVs were subjected to kinase reaction or left untreated. After the kinase reaction, vesicles were washed to remove unreacted components or damaged vesicles. They were incubated with FGF2-GFP for 1 h and imaged with confocal microscope to access the conversion of PI(4)P to PI(4,5)P$_2$. Representative confocal images of FGF2-GFP binding to GUVs and their binding intensity profile with and without kinase reaction are shown ($n = 3$). For statistical tests, ordinary one-way ANOVA was performed in Prism. Not significant (ns) $P > 0.5$, ***$P < 0.001$, ****$P < 0.0001$. Data distribution was assumed to be normal. Source data are provided as a Source Data file.

which, apparent binding intensity recovered to normal when compared to non-treated PI(4,5)P$_2$ vesicle (Fig. 2B, condition a, light red bar versus c, light red bar). For LUVs containing PI(4)P, treated with PIP5K1C, Mg$^{2+}$, and ATP, when directly analyzed for FGF2-GFP binding efficiencies, similar levels were observed compared to LUVs containing PI(4,5)P$_2$ (Fig. 2B, condition c, grey bar versus d, grey bar). Most importantly, when LUVs containing PI(4)P, treated with PIP5K1C, Mg$^{2+}$ and ATP were subjected to the disintegration/reformation procedure, FGF2-GFP binding efficiencies did not went up as observed for PI(4,5)P$_2$ LUVs (Fig. 2B, condition d, light red bar). A comparison of FGF2 binding efficiencies with LUVs containing different concentrations of PI(4,5)P$_2$ (Fig. 2C) indicated that such LUVs had a surface concentration of enzymatically produced PI(4,5)P$_2$ between 2 and 3 mol%. Compared to the starting condition of 5 mol%, this indicated that the disintegration/reformation procedure indeed caused dilution of PI(4,5)P$_2$ into the inner leaflet. This validated our approach with LUVs formed with PI(4)P followed by enzymatic conversion into PI(4,5)P$_2$ being characterized by an asymmetric transbilayer distribution of PI(4,5)P$_2$ with at least the vast majority of this membrane lipid being present in the outer leaflet of LUVs.

## PIP5K1C mediated conversion of PI(4)P into PI(4,5)P$_2$ on the membrane surface of both LUVs and GUVs with a plasma membrane-like lipid composition

The basic approach to generate liposomes with an asymmetric transbilayer distribution of PI(4,5)P$_2$ was based on the PIP5K1C-mediated conversion of PI(4)P into PI(4,5)P$_2$ at the outer leaflet of LUVs and GUVs. As a first step, using a simple bulk lipid composition consisting primarily of phosphatidylcholine, we established the methodology and validated the production of asymmetric PI(4,5)P$_2$ liposomes (Figs. 1 and 2). To prepare for functional experiments such as FGF2-dependent membrane pore formation, we aimed at making the lipid composition more complex, including phosphatidylethanolamine, phosphatidylserine, phosphatidylinositol, cholesterol, and sphingomyelin, mimicking the lipid composition of plasma membranes. We first produced LUVs with a plasma membrane-like lipid composition containing either PI(4)P or PI(4,5)P$_2$ (Fig. 3A). To quantify the presence of PI(4,5)P$_2$, in addition to FGF2-GFP, we used a FGF2-Halo fusion protein. On the one hand, FGF2-Halo fusion proteins have the advantage to reduce liposomes clustering; however, compared to FGF2-GFP, binding to PI(4,5)P$_2$ is weaker due to the acidic isoelectric point of the Halo domain[13]. By using both FGF2-GFP and FGF2-Halo fusion proteins in this study, we also made sure that the observed phenomena were not specific to a certain PI(4,5)P$_2$ sensor protein. To test the efficiency of kinase reaction, we measured FGF2-Halo binding using the biochemical sedimentation assay. The binding efficiency of FGF2-Halo towards PI(4,5)P$_2$ containing LUVs (green bars) was set as a reference point (Fig. 3A, green bar, condition 1). A substantial reduction in FGF2-Halo binding efficiency was observed for LUVs with a plasma

membrane-like lipid composition containing PI(4)P (Fig. 3A, purple bars, condition 2). As shown in Fig. 3A (condition 3 and 4), a binding efficiency of FGF2-Halo comparable to PI(4,5)P$_2$ LUVs could be recovered when PI(4)P containing LUVs were incubated with PIP5K1C, Mg$^{2+}$, and ATP, converting PI(4)P into PI(4,5)P$_2$. By contrast, under mock conditions (Fig. 3A, purple bars, condition 5), FGF2-Halo failed to bind to liposomes that were treated with ATP and Mg$^{2+}$ ions in the absence of PIP5K1C. Similar results were obtained with GUVs containing a plasma membrane-like lipid composition supplemented with either PI(4,5)P$_2$ or PI(4)P (Fig. 3B, C). Using FGF2-GFP as a PI(4,5)P$_2$ sensor, we observed efficient binding to GUVs containing PI(4,5)P$_2$ (Fig. 3B, control) and absence of binding when GUVs contained PI(4)P (Fig. 3C, control). Of note, conversion of PI(4)P into PI(4,5)P$_2$ restored FGF2-GFP binding to GUVs (Fig. 3B, C, kinase reaction). A similar trend was observed when FGF2-Halo was used in GUV experiments. The combined experiments shown in Figs. 1, 2, and 3 established that conversion of PI(4)P into PI(4,5)P$_2$ mediated by PIP5K1C works both in a simple PC background (Figs. 1 and 2) and in a more complex lipid background mimicking the plasma membrane lipid composition (Fig. 3). In addition, incubations of PIP5K1C in the presence of ATP and Mg$^{2+}$ ions did not compromise membrane integrity of GUVs throughout the experiment as indicated by the exclusion of a small fluorescent tracer, Alexa-647 (Fig. 3D, E as well as Supplementary Movies 1 and 2). Similar to the experiments shown in Figs. 1 and 2, this indicates that PIP5K1C acted on PI(4)P exclusively from the outside, producing PI(4,5)P$_2$ in the outer leaflet of GUVs only. Hence, alike the experiments shown in Figs. 1 and 2, this procedure produced GUVs with an asymmetric transbilayer distribution of PI(4,5)P$_2$.

## PI(4,5)P$_2$ asymmetry facilitates FGF2-dependent membrane pore formation in reconstitution experiments

To test the hypothesis as to whether the kinetics of FGF2 membrane pore formation are affected by the transbilayer distribution of PI(4,5)P$_2$, we used GUVs with a plasma membrane-like lipid composition in combination with previously established imaging assays to monitor FGF2-induced membrane pore formation[9,12,14]. Three experimental conditions were compared with (i) GUVs with a symmetric transbilayer distribution, (ii) GUVs with an asymmetric transbilayer distribution of PI(4,5)P$_2$, and (iii) GUVs with a symmetric transbilayer distribution treated with PIP5K1C, ATP, and Mg$^{2+}$ ions as a control (Fig. 4). In Fig. 4A, examples of still images of GUVs are shown at an early time point (20 to 25 min) of incubation with FGF2-GFP. Along with a fluorescently labeled membrane lipid (RhPE), FGF2-GFP binding to the membrane surfaces of GUVs was visible for all three conditions. Importantly, at these time points, membrane pore formation was not detectable as indicated by the luminal exclusion of the small fluorescent tracer, Alexa-647. In Fig. 4B, examples are given for individual GUVs from later time points along with the incubation time that led to FGF2-dependent membrane pore formation, indicated by luminal

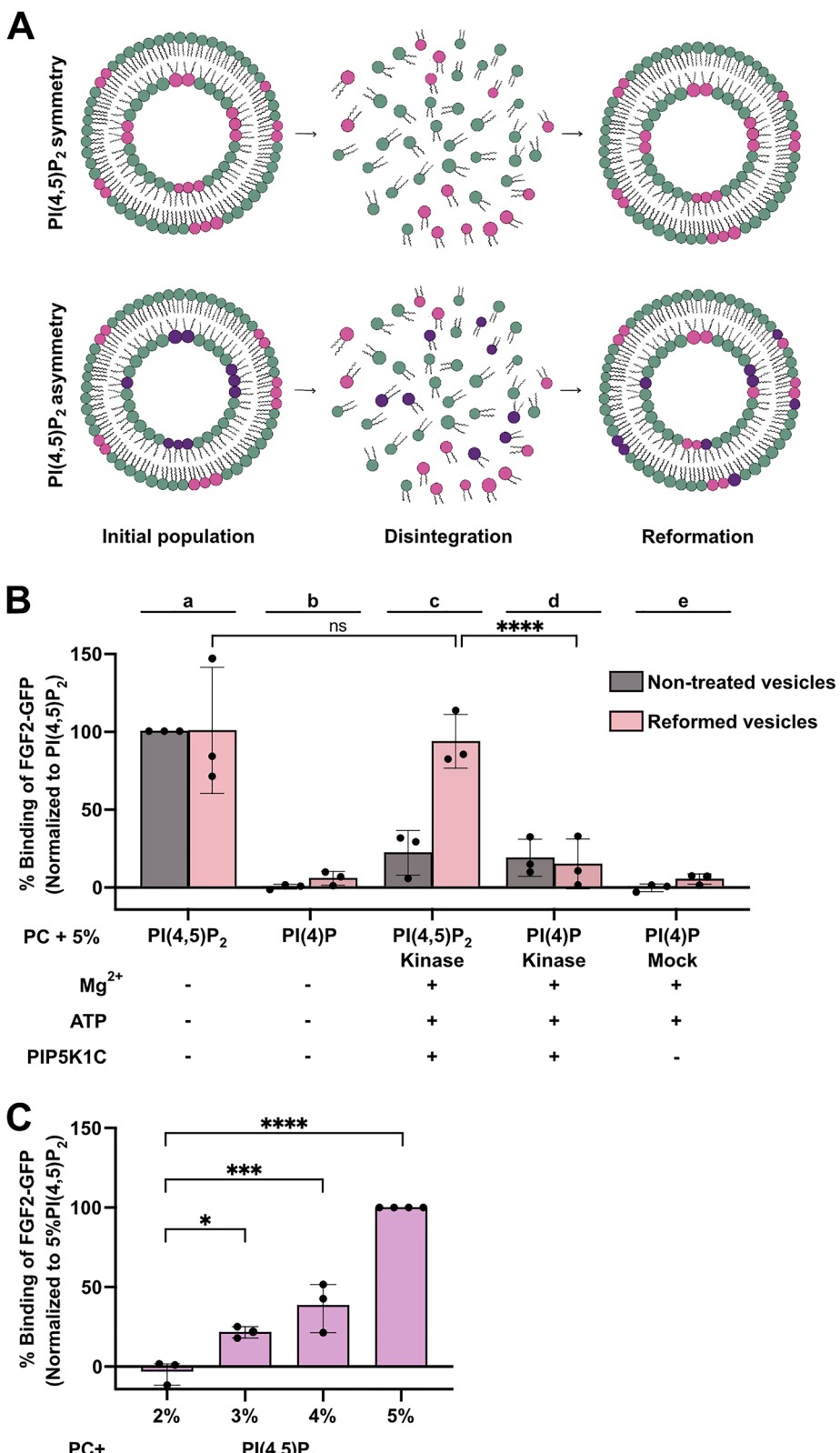

penetration of the small fluorescent tracer molecule (Alexa-647). For the given examples, this event occurred after about 2 h for the GUV with a symmetric transbilayer distribution, after 40 min for the GUV with an asymmetric transbilayer distribution of PI(4,5)P$_2$ and after 1 h and 27 min for the GUV with a symmetric transbilayer distribution treated with PIP5K1C, ATP and Mg$^{2+}$ ions as a control (Fig. 4B). The Supplementary Movies 3, 4 and 5 cover the incubation time for all

three of these conditions, visualizing the dynamics of FGF2-dependenet pore formation at different time points depending on the experimental conditions described above. In some cases, GUVs appeared to undergo phase separation, displaying regions in which FGF2-GFP binding was enhanced. However, this phenomenon was not consistently observed across all experiments with those variations not having an impact on FGF2 membrane pore formation. Nevertheless,

**Fig. 2 | Testing transbilayer PI(4,5)P$_2$ asymmetry in large unilamellar vesicles.**
**A** Schematic representation of asymmetry test using vesicle disintegration/reformation method. Vesicles were disintegrated with 40% MeOH in CHCl$_3$ and then reconstructed to assess their binding towards FGF2-GFP (adapted from ref. 56).
**B** Quantification of FGF2-GFP binding to PC Large Unilamellar Vesicles (LUVs) using FACS flow cytometry. Vesicles were incubated with FGF2-GFP for 30 min, and unbound protein was removed by centrifugation before analysis. Binding was assessed for LUVs with 5 mol% PI(4,5)P$_2$ (sub-panel a), 5 mol% PI(4)P (sub-panel b), 5 mol% PI(4,5)P$_2$ + Mg$^{2+}$ + ATP + PIP5K1C (sub-panel c), 5 mol% PI(4)P + Mg$^{2+}$ + ATP + PIP5K1C (sub-panel d), 5 mol% PI(4)P + Mg$^{2+}$ + ATP (sub-panel e). It is important to note that the vesicles were washed after the kinase reaction to remove unreacted components or damaged vesicles. Gray bars represent initial/non-treated population of LUVs and light red bar represent reformed LUVs. Each dot represents a biological replicate; values are the mean of 2 technical replicates.

For each experiment, the average signal from non-treated/initial 5 mol% PI(4,5)P$_2$ LUV population was set to 100%, and all values were normalized accordingly. Mean values with standard deviation (S.D.) for 3 biological replicates are presented.
**C** Analysis of FGF2-GFP binding to PC LUVs with increasing mol% PI(4,5)P$_2$ using FACS flow cytometry. Vesicles were incubated with FGF2-GFP for 30 min, and unbound protein was removed by centrifugation before analysis. Each dot represents a biological replicate; values are the mean of 2–3 technical replicates. For each experiment, average signal from 5 mol% PI(4,5)P$_2$ was set to 100%, and all values were normalized accordingly. Mean values with S.D. for 3–4 biological replicates are presented. For statistical tests, ordinary one-way ANOVA was performed in Prism. Not significant (ns) $P > 0.5$, * $P < 0.05$, *** $P < 0.001$, **** $P < 0.0001$. Data distribution was assumed to be normal. Source data are provided as a Source Data file.

phase separation might be of relevance for future studies as we have hypothesized the FGF2 membrane translocation machinery to be housed in liquid-ordered nanodomains[26].

To test the significance of the observed differences depicted in Fig. 4A, B, we recorded full movies of the time intervals required for FGF2-dependent membrane pore formation for 8 individual GUVs per condition, derived from 4 to 7 independent experiments. These measurements were conducted for each of the three experimental conditions described (Fig. 4C). This analysis revealed a statistically significant shortening of time intervals required for FGF2-dependent membrane pore formation for GUVs with an asymmetric transbilayer distribution of PI(4,5)P$_2$ (mean value of about 45 min) compared with those having a symmetric distribution of membrane lipids (mean value of about 95 min). By contrast, the time intervals for FGF2-dependent membrane pore formation measured for GUVs with a symmetric transbilayer distribution and those additionally treated with PIP5K1C, ATP and Mg$^{2+}$ ions did not differ with a mean value of about 105 min for the latter (Fig. 4C). Of note, the total efficiency of FGF2-dependent membrane pore formation observed as the percentage of GUVs containing the Alexa-647 tracer at the end of the measurements did not differ between GUVs with a symmetric versus those having an asymmetric distribution of PI(4,5)P$_2$ (Fig. 4D). In addition, we tested PI(4)P-containing GUVs which were left untreated or treated with PIP5K1C and Mg$^{2+}$ in the absence of ATP. Under these conditions, membrane pore formation was undetectable, indicating that the presence of PIP5K1C and Mg$^{2+}$ ions do not affect this process in a pleiotropic manner.

To further corroborate the effect of PI(4,5)P$_2$ transbilayer asymmetry on the kinetics of FGF2-dependent membrane pore formation, we performed a bulk quantification measuring the percentage of GUVs filled with the Alexa-647 tracer as a read out of FGF2 induced pore formation (Fig. 5). This analysis was carried out at 0, 40, 90 min on the various types of GUVs and experimental conditions indicated. At timepoint 0, FGF2-GFP induced pore formation was almost undetectable under all conditions (Fig. 5A). At 40 min of incubation, a substantial fraction of GUVs containing an asymmetric transbilayer distribution of PI(4,5)P$_2$ displayed pore formation (Fig. 5B). Only following longer incubation with a time point of 90 min, GUVs with a symmetric distribution of PI(4,5)P$_2$ reached comparable levels of membrane pore formation (Fig. 5C). By contrast, PI(4)P-containing GUVs did not display membrane pore formation throughout the entire time course of these experiments. The findings documented in Fig. 5 corroborate transbilayer asymmetry of PI(4,5)P$_2$ to represent an important parameter for the thermodynamic properties of this system, governing the kinetics of FGF2-induced membrane pore formation.

## Disturbing plasma membrane transbilayer asymmetry of PI(4,5)P$_2$ in intact cells

To challenge the results from the biochemical reconstitution experiments shown in Figs. 1–5 and the accompanying

Supplementary Movies 3–5, we went further to test as to whether plasma membrane transbilayer asymmetry of PI(4,5)P$_2$ in intact cells plays a role in unconventional secretion of FGF2. We established a procedure to disturb the asymmetric distribution of PI(4,5)P$_2$ in intact cells that was inspired by the delivery of functionalized phosphoinositides to cells[36–38]. As a control, we disrupted asymmetric distribution of phosphatidylserine (PS), an abundant acidic membrane lipid that, similar to phosphoinositides, is localized exclusively to the cytoplasmic leaflet of the plasma membrane. As illustrated in Fig. 6A, PI(4,5)P$_2$ and PS used as a control were added to cells in an aqueous solution, most likely forming micelles (PI(4,5)P$_2$) or liposomal vesicles (PS). We reasoned cellular delivery would occur at the outer plasma membrane leaflet, compromising transbilayer asymmetry of the endogenous counterparts of both of those membrane lipids that under normal conditions localize exclusively to the inner plasma membrane leaflet in living cells. Delivery was experimentally confirmed by confocal imaging using a recombinant PH domain-based fluorescent sensor for PI(4,5)P$_2$ (PHx2-Halo[13]); and a fluorescent variant of Annexin V for PS. As shown in Fig. 6B, as opposed to mock conditions, both PI(4,5)P$_2$ (left-hand panels) and PS (right-hand panels) were readily detectable by the recombinant fluorescent sensors that were added to cells from outside, detecting these lipids in the outer plasma membrane leaflet. Based on our calculations of PI(4,5)P$_2$ levels for individual cells (see "Materials and Methods"), we added a threefold excess of PI(4,5)P$_2$. We reasoned that this concentration would effectively disrupt PI(4,5)P$_2$ transbilayer asymmetry, which is consistent with the signal observed using the PHX2-Halo protein.

## A quantitative assay measuring an acute wave of FGF2 secretion from cells

We have previously established a number of quantitative assays measuring FGF2 transport into the extracellular space based upon, for example, cell surface biotinylation as well as confocal and TIRF microscopy[6,8–11,16,24,25,39,40]. These experimental systems were designed to quantify FGF2 on cell surfaces at equilibrium, with the experiment initiated by doxycycline-dependent induction of FGF2 expression and cell surface FGF2 quantified after the system reached equilibrium. In the context of the current study, we aimed at developing an experimental system based upon a single cell analysis using confocal microscopy, quantifying an acute wave of FGF2 translocation to cell surfaces within a short time interval in the range of minutes (Fig. 7). Cells were treated for 16 h with doxycycline to induce FGF2-GFP expression with an equilibrium distribution between the cytoplasm, the nucleus and the cell surface. While the first two subcellular localizations were directly observable by GFP fluorescence, cell surface FGF2-GFP was detected on intact cells using fluorescently labeled anti-GFP antibodies (Fig. 7A). In a second step, cells were treated with heparin molecules containing high affinity binding sites for FGF2 to wash away and remove FGF2-GFP from cell surfaces (Fig. 7B). The

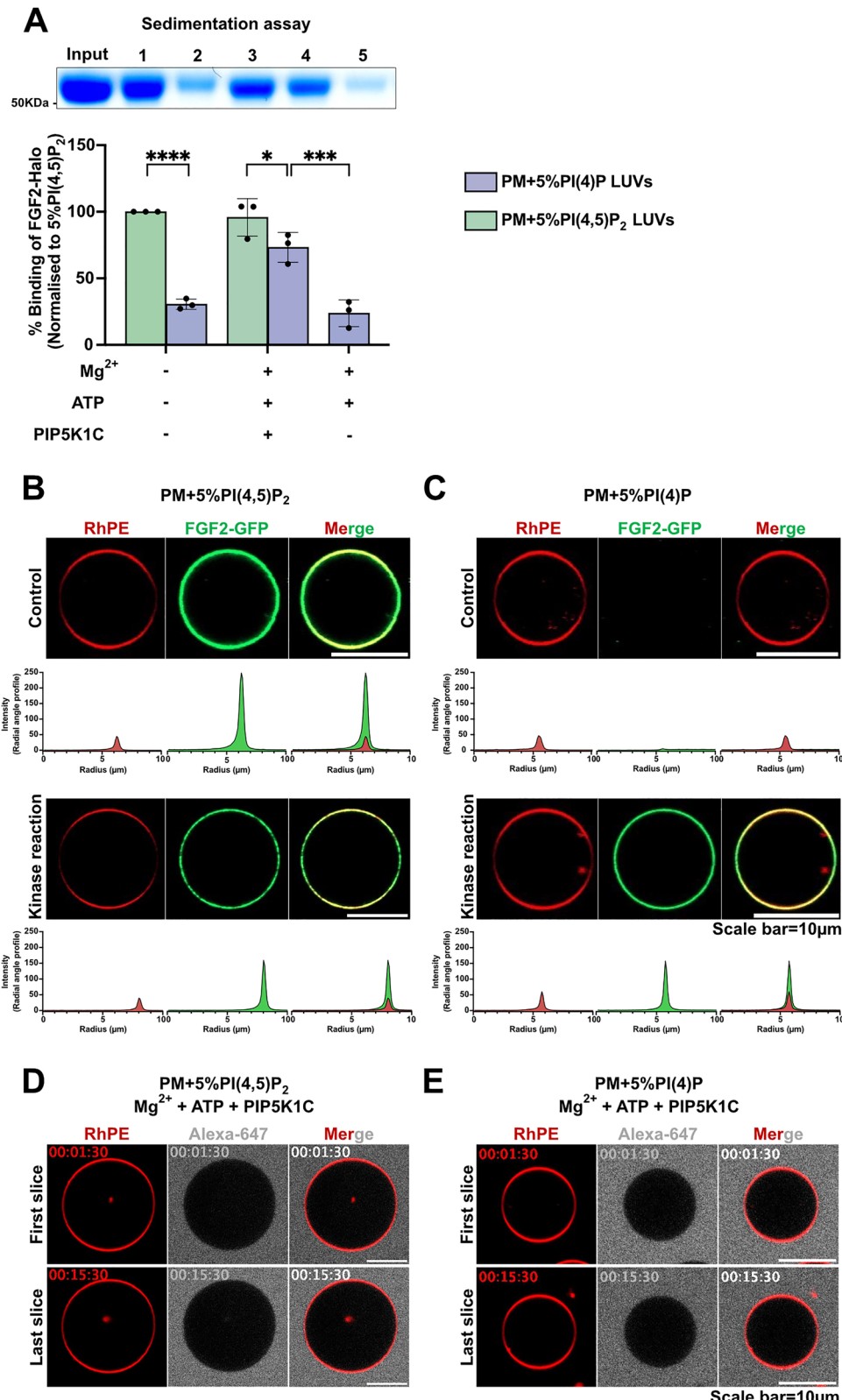

condition right after removal of FGF2-GFP from cell surfaces was defined as timepoint 0. Subsequently, as shown in Fig. 7C, a synchronized acute wave of FGF2-GFP secretion could be observed by the reappearance of FGF2-GFP on cell surfaces. With the example given in Fig. 7, reappearance of FGF2-GFP on cell surfaces was substantial already after a 30 min interval of chase. As demonstrated by confocal imaging and a quantitative single cell analysis (Fig. 7D), this

experimental setup work in a robust manner with an efficient removal of FGF2-GFP from cell surfaces by a stringent heparin wash (Fig. 7D, condition a versus b) and an efficient recovery of the cell surface population of FGF2-GFP within 30 min of incubation (Fig. 7D, condition c), representing an acute wave of FGF2 secretion into the extracellular space.

**Fig. 3 | Generation of PI(4,5)P₂ asymmetric vesicles with plasma membrane-like lipid composition. A** Large unilamellar vesicles (LUVs) were prepared with 5 mol% PI(4,5)P₂ or 5 mol% PI(4)P in plasma membrane (PM) like background. These LUVs were subjected to kinase reaction or mock reaction (without PIP5K1C) or left untreated. After the reaction, vesicles were washed and incubated with FGF2-Halo for 3 h, and unbound protein was removed by centrifugation before analysis using a sedimentation assay. The bound fraction of FGF2-Halo was analyzed by SDS-PAGE and detected with Coomassie staining. The gel was imaged using EPSON 4870 scanner. Data were normalized to 5 mol% PI(4,5)P₂ within each experiment. Mean values with standard deviation for 3 biological replicates are presented. For statistical tests, ordinary one-way ANOVA was performed in Prism. Not significant (ns) $P > 0.5$, $* P < 0.05$, $*** P < 0.001$, $**** P < 0.0001$. Data distribution was assumed to be normal. **B** Giant unilamellar vesicles (GUVs) were prepared with 5 mol% PI(4,5)P₂ or **C** 5 mol% PI(4)P along with membrane marker Rhodamine-PE, in PM like background. GUVs were subjected to kinase reaction or left untreated. After the kinase reaction, vesicles were washed to remove unreacted components or damaged vesicles. They were incubated with FGF2-GFP for 1 h and imaged with confocal microscope to access the enzymatic conversion of PI(4)P to PI(4,5)P₂. Representative confocal images of FGF2-GFP binding to GUVs and their binding intensity profile with and without kinase reaction are shown ($n = 3$). **D** To test the asymmetry in GUVs, vesicles with 5 mol% PI(4,5)P₂ or **E** 5 mol% PI(4)P along with membrane marker Rhodamine-PE, in PM like background, were immobilized on glass bottom ibidi chambers. Vesicles were monitored after the addition of Mg²⁺ + ATP + PIP5K1C and small tracer dye Alexa-647 in real time for 15 min using confocal microscope. Each time frame is separated by 1 min time. First slice was recorded at 1.5 min and last time slice was recoded at 15 min. Representative confocal images of GUVs with kinase reaction remains intact during the entire course of kinase reaction. Source data are provided as a Source Data file.

## Transbilayer asymmetry of PI(4,5)P₂ in plasma membranes is required for FGF2 secretion from cells

Using the experimental setup described in Fig. 7, we tested as to whether disturbance of the asymmetric distribution of PI(4,5)P₂ in the plasma membranes of cells directly affects the efficiency of FGF2 membrane translocation into the extracellular space (Fig. 8). Four conditions were compared to quantify FGF2 secretion with (i) untreated cells, (ii) mock-treated cells, (iii) cells treated with exogenous PI(4,5)P₂ and (iv) cells treated with exogenous PS, using the procedures established in the experiments shown in Fig. 6. In Fig. 8A, representative example images for each of the four experimental conditions are shown. For untreated and mock-treated as well as PS-treated cells, efficient reappearance of FGF2-GFP on cell surfaces could be observed within 30 min after the heparin wash, demonstrating normal FGF2-GFP transport efficiencies into the extracellular space under these conditions. By contrast, in cells treated with exogenous PI(4,5)P₂ to disturb transbilayer asymmetry, FGF2-GFP failed to reappear at the cell surface, demonstrating FGF2 membrane translocation to be blocked under this experimental condition. Quantification along with statistics comparing the four experimental conditions mentioned above were done with single cell resolution and are shown in Fig. 8B. At timepoint 0 (Fig. 8B, left-hand side), FGF2-GFP was hardly detectable on cell surfaces, demonstrating the effectiveness of the heparin wash, marking the start condition of the experiment. After 30 min of incubation (Fig. 8B, right-hand side), substantial amounts of FGF2-GFP on cell surfaces could be detected for untreated, mock-treated, and PS-treated cells without any statistically significant differences. By contrast, cells treated with exogenous PI(4,5)P₂ were characterized by very low levels of FGF2-GFP on cell surfaces similar to the one detected at 0 min of incubation. The observed differences for PI(4,5)P₂-treated cells compared to mock- and PS-treated cells were statistically highly significant, demonstrating the specific effect of the disruption of PI(4,5)P₂ transbilayer asymmetry on FGF2 membrane translocation to cell surfaces.

In order to introduce an even higher degree of stringency regarding negative controls such as PS (Fig. 8), we also disrupted the transbilayer asymmetry of other phosphoinositides by adding to cell surfaces (i) PI(4)P, (ii) PI(3,4)P₂, and (iii) PI(3,4,5)P₃ (Fig. 9). Similar to the PS control, these experiments revealed PI(4)P addition to cell surfaces having no inhibitory effect on FGF2 secretion. By contrast, the addition of either PI(3,4)P₂ or PI(3,4,5)P₃ did cause inhibition of FGF2 secretion in a similar manner as observed for PI(4,5)P₂ (Fig. 9A, B). These observations are interesting as we showed in previous studies that, albeit with lower efficiency, FGF2 does not only bind to PI(4,5)P₂ but also to PI(3,4)P₂ and PI(3,4,5)P₃, a clear difference compared to PS and PI(4)P that were demonstrated not to interact with FGF2[6,34]. To further challenge the findings documented in Figs. 8 and 9, we analyzed the lifetime of exogenously added PI(4,5)P₂ on cell surfaces during the experiments, using the procedures described in Figs. 6 and 7. As shown in Fig. 10A, starting at about 40 min following addition of PI(4,5)P₂, the cell surface levels of PI(4,5)P₂ started to decrease and were hardly detectable at 60 min of incubation. This suggested that, at time points later than our reference point of 30 min, cells could re-establish PI(4,5)P₂ asymmetry in their plasma membranes, presumably by endocytosis and/or perhaps flippase-mediated removal of exogenously added PI(4,5)P₂. Intriguingly, as shown in Fig. 10B, FGF2-GFP translocation to cell surfaces recovered when exogenously added PI(4,5)P₂ disappeared from cell surfaces at late time points of 40 to 60 min. These findings corroborate our conclusion that a failure of FGF2-GFP to reach cell surfaces at 30 min of incubation is due to the loss of PI(4,5)P₂ asymmetry, exerted by the addition of exogenous PI(4,5)P₂ from outside cells. In conclusion, transbilayer asymmetry of PI(4,5)P₂ in plasma membranes is a key parameter enabling efficient FGF2 membrane translocation into the extracellular space.

## Discussion

A general hallmark of biological membranes from a broad range of organisms is their structural organization in laterally partitioned nanodomains as well as their transbilayer asymmetry of membrane lipid distribution between the two leaflets[41–43]. For example, sphingomyelin is known to be mainly restricted to the outer plasma membrane leaflet[41]. On the other hand, phosphoinositides and phosphatidylserine (PS) are found almost exclusively in the inner plasma membrane leaflet[41]. To preserve this membrane lipid transbilayer asymmetry, cells invest significant amounts of energy through ATP-dependent flippases and floppases that actively maintain the lipid distribution between the two leaflets of the plasma membrane[44,45]. While the structure-function relationship of laterally partitioned nanodomains containing the molecular machineries for a diverse set of physiological functions has been studied in great detail, much less is known about the functional implications of the asymmetric transbilayer distribution of membrane lipids in biological membranes[41,46]. Nevertheless, it is widely assumed that this asymmetry is of broad relevance for proper cellular functions and that the disturbance of plasma membrane lipid transbilayer asymmetry can lead to malfunctions and potentially disease.

As of this day, only for a few cases, a causally determined relationship between membrane lipid transbilayer asymmetry and cellular functions has been established. An example is the regulation of the activity of membrane proteins depending on an asymmetric distribution of membrane lipids, being a critical parameter for cellular health[47]. Furthermore, membrane asymmetry of PS localized in the inner plasma membrane leaflet in benign cells is a prerequisite for signaling during apoptosis, a process that causes a global loss of membrane asymmetry by which PS is exposed on the cell surface and serves as an "eat me" signal to phagocytes, removing deranged cells that may harm the entire organism[48]. Another example is a recent study demonstrating loss of membrane asymmetry to affect cellular energetics, shifting from anabolic to catabolic pathways and slowing down cell

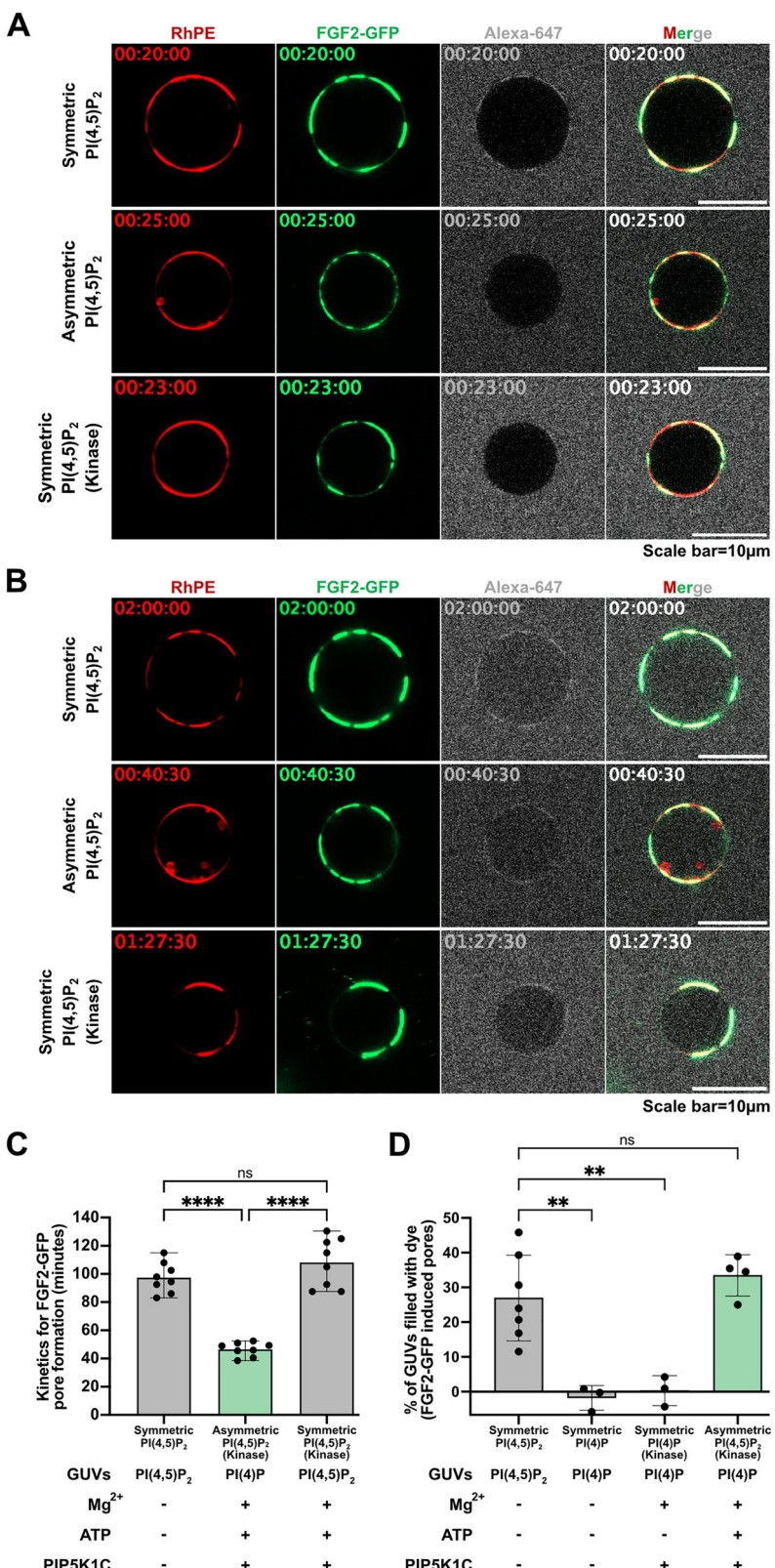

division[49]. Other examples relate to a temporary loss of plasma membrane lipid asymmetry that plays a critical role in maintaining normal physiological functions. For instance, recent studies revealed plasma membrane SM asymmetry to be disrupted during lysosomal damage. Calcium release from damaged lysosomes was shown to activate a SM scramblase, flipping SM to the cytosolic leaflet, where it is converted into ceramide[50]. This constitutes a signal that leads to lysosomal repair mediated by an ESCRT-independent pathway. Similarly, a loss of PE asymmetry has been observed during cytokinesis[51]. Although the mechanisms underlying this process are not yet fully understood, this phenomenon is crucial for effective cell division[52]. During cytokinesis, the two membrane leaflets are coupled, with PE being enriched in the outer leaflet of the cleavage furrow, alongside SM and cholesterol. By contrast, phosphatidylinositol PI(4,5)P2 is enriched

**Fig. 4 | FGF2 translocation kinetics depend on the asymmetric transbilayer distribution of PI(4,5)P₂ based on individual GUV analyses.** Giant unilamellar vesicles (GUVs) were prepared with 5 mol% PI(4,5)P₂ or 5 mol% PI(4)P along with membrane marker Rhodamine-PE, in PM like background. Small trace dye Alexa-647 was added to record the event of pore formation for kinetic measurement. Three conditions were compared for FGF2 translocation kinetic study; GUVs with symmetric PI(4,5)P₂, asymmetric PI(4,5)P₂, and symmetric PI(4,5)P₂ with Mg²⁺ + ATP + PIP5K1C as a control. Following the kinase reaction, vesicles were washed to remove unreacted components or damaged vesicles. After addition of FGF2-GFP, time was marked as 0 min and vesicle was given 10−20 min to immobilize and imaging was started at 20−25 min till 01:30−02:00 h. **A** First recorded time frame for PM vesicles with symmetric or asymmetric PI(4,5)P₂ or symmetric vesicles with reaction ingredients. **B** Still frame at the real time of pore formation for PM vesicles with symmetric or asymmetric PI(4,5)P₂ or symmetric vesicles with reaction ingredients. **C** Quantification and statistical analysis of pore formation kinetics for FGF2-GFP in symmetric (grey bars), asymmetric PI(4,5)P₂ (light green bars), and symmetric vesicles with reaction ingredients (grey bars). Each dot

represents a single GUV pore formation in real time coming from eight individual GUVs. **D** To access the bulk percentage of GUVs with dye inside were analyzed for PI(4,5)P₂, PI(4)P, PI(4)P GUVs subjected to kinase reaction in the presence and absence of ATP. After the kinase reaction, vesicles were washed to remove unreacted components or damaged vesicles. Quantification and statistical analysis of FGF2-GFP dependent pore formation is presented. From each condition percentage of GUVs with dye inside in the absence of protein were subtracted to access FGF2-dependent pore formation. Each dot represents percentage of GUVs with membrane pores with a ratio of luminal versus exterior Alexa647 tracer ≥0.6. To compute percentage of GUVs filled with Alexa647 dye, in each independent experiment, 20−135 GUVs were examined. For Symmetric PI(4,5)P₂ n = 7, Symmetric PI(4)P n = 3, Symmetric PI(4)P (Kinase) n = 3, Asymmetric PI(4,5)P₂ n = 4. Mean values with S.D. for biological replicates are presented. Symmetric GUVs are represented by grey bars and asymmetric GUVs are presented by light green bars. For statistical test, ordinary one-way ANOVA was performed in Prism. Not significant (ns) $P > 0.5$, ** $P < 0.01$, **** $P < 0.0001$. Data distribution was assumed to be normal. Source data are provided as a Source Data file.

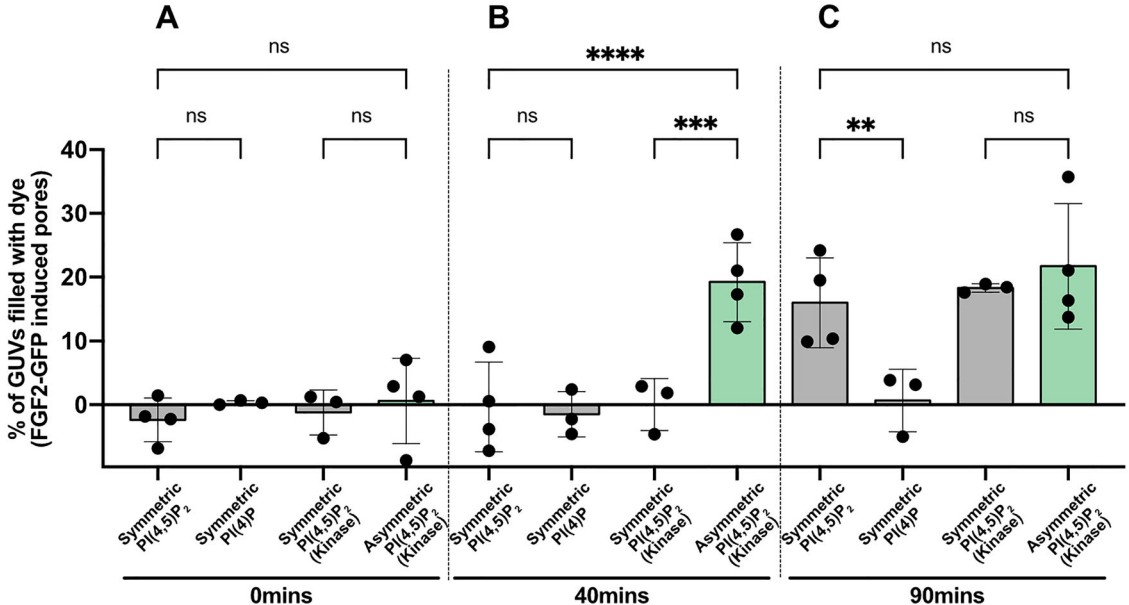

**Fig. 5 | FGF2 translocation kinetics depend on the asymmetric transbilayer distribution of PI(4,5)P₂ based on bulk quantification of GUVs.** Giant unilamellar vesicles (GUVs) were prepared with 5 mol% PI(4,5)P₂ or 5 mol% PI(4)P along with membrane marker Rhodamine-PE, in PM like background. GUVs were immobilized in glass bottom ibidi chambers. Four conditions were compared for FGF2 translocation kinetic study; GUVs with symmetric PI(4,5)P₂, symmetric PI(4,5)P₂ with Mg²⁺ + ATP + PIP5K1C as a control, symmetric PI(4)P, and asymmetric PI(4,5)P₂. Following the kinase reaction, vesicles were washed to remove unreacted components. After addition of FGF2-GFP, and a small tracer dye Alexa-647, time was marked as 0 min. GUVs were imaged at **A** 0 min, **B** 40 min, and **C** 90 min. It is important to mention that for each time point, 25−30 min were taken to take sufficient images for bulk quantification. From each condition percentage of GUVs

with dye inside in the absence of protein were subtracted to access FGF2-dependent pore formation. Symmetric GUVs are represented by grey bars and asymmetric GUVs are presented by light green bars. Each dot represents percentage of GUVs with membrane pores with a ratio of luminal versus exterior Alexa647 tracer ≥0.6. To compute percentage of GUVs filled with Alexa647 dye, in each independent experiment 19−121 GUVs were examined. For Symmetric PI(4,5)P₂ n = 4, Symmetric PI(4)P n = 3, Symmetric PI(4,5)P₂ (Kinase) n = 3, Asymmetric PI(4,5)P₂ n = 4. Mean values with S.D. for biological replicates are presented. For statistical test, ordinary one-way ANOVA was performed in Prism. Not significant (ns) $P > 0.5$, ** $P < 0.01$, *** $P < 0.001$, **** $P < 0.0001$. Data distribution was assumed to be normal. Source data are provided as a Source Data file.

in the inner leaflet. Disrupting the transbilayer distribution of any of these lipids was shown to impair cell division[53,54].

The strong coupling of membrane lipids such as SM, cholesterol, and PI(4,5)P₂ between the two leaflets during cytokinesis shows intriguing similarities to the mechanism of FGF2 membrane translocation during unconventional secretion. Cholesterol has been demonstrated to tune PI(4,5)P₂-dependent recruitment of FGF2 at the inner plasma membrane leaflet, resulting in substantial changes in the efficiency of unconventional secretion of FGF2 when plasma membrane cholesterol levels are manipulated[13,26]. Due to the intimate relationship between cholesterol and SM in the plasma membrane, it is likely that SM levels are also important for efficient secretion of FGF2. Along with the

observation of Glypican-1 (GPC1), a GPI-anchored cell surface protein localized in cholesterol-rich ordered domains, being the primary heparan sulfate proteoglycan driving FGF2 secretion[3,15], the hypothesis of nanodomains containing the membrane translocation machinery for FGF2 was inspired[26]. Following the capturing of FGF2 oligomers by GPC1 and their removal at the outer leaflet, the membrane pore may close spontaneously. This process has been imaged by single molecule TIRF microscopy in real time, demonstrating FGF2 membrane translocation events to be rather fast with average time intervals of about 200 ms[4,25].

Building on these insights, the current study provides, to our knowledge, the direct demonstration in a physiological context that

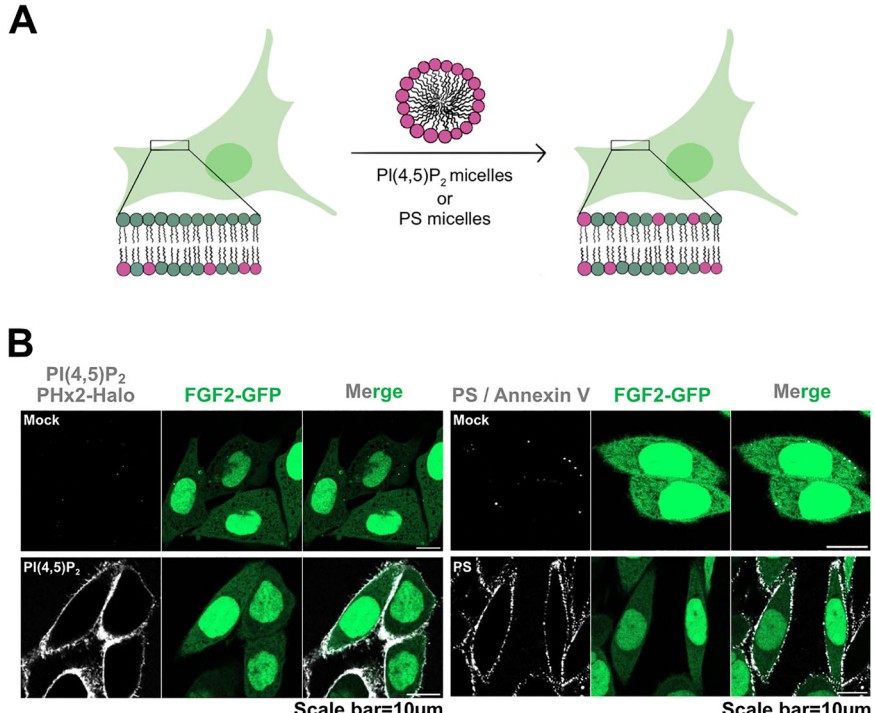

**Fig. 6 | Disruption of cellular transbilayer asymmetry of PI(4,5)P₂ or PS.**
**A** Schematic representation of the disruption of plasma membrane PI(4,5)P₂ or PS transbilayer asymmetry using PI(4,5)P₂ or PS micelles/vesicles (adapted from ref. 56). **B** Representative confocal images of CHO-K1 cells expressing FGF2-GFP in doxycycline-dependent manner, with disrupted PI(4,5)P₂ or PS asymmetry (n = 3). On left of (**B**), are fixed cells stained with high affinity PI(4,5)P₂ binding protein PHX2-Halo (40 nM). The Halo tag was visualized using Halo ligand Alexa 660

(75 nM). Cells were fixed using 3% PFA in PBS for 13 min at room temperature. PI(4,5)P₂ signal was observed only upon addition of PI(4,5)P₂ micelles, confirming the successful disruption of PI(4,5)P₂ asymmetry. On right of (**B**) are live cells stained with PS binding protein AnnexinV-Alexa647 in presence of 2 mM Ca²⁺ in Live cell imaging solution. PS signal was observed only upon addition of PS micelles, confirming the successful disruption of PS asymmetry. Source data are provided as a Source Data file.

transbilayer asymmetry of PI(4,5)P₂ is a functional determinant of membrane pore formation. We demonstrate an asymmetric distribution of PI(4,5)P₂ to facilitate membrane pore formation mediated by PI(4,5)P₂-dependent FGF2 oligomerization and to be essential for FGF2 secretion from cells. This finding not only supports a previously hypothesized role for lipid asymmetry in membrane remodeling but also establishes a direct causal link in living cells. Therefore, we reveal PI(4,5)P₂ transbilayer asymmetry to represent an active and tunable mechanism that controls FGF2-induced membrane pore formation. This observation therefore, represents a clear example of the importance of transbilayer asymmetry of membrane lipids for a fundamental process in cell biology.

In terms of the molecular mechanism underlying FGF2 membrane translocation into the extracellular space, we propose the local accumulation of PI(4,5)P₂ molecules underneath FGF2 oligomers recruited at the inner plasma membrane leaflet to produce a steep electrochemical gradient across the membrane that facilitates the opening of FGF2-induced lipidic membrane pores[31]. Furthermore, with PI(4,5)P₂ representing a non-bilayer lipid and due to its asymmetric distribution, we propose FGF2 oligomer-dependent accumulation of PI(4,5)P₂ in a locally constrained manner to cause a massive amount of membrane stress. This in turn may compromise the structural integrity of the regular bilayer, facilitating the opening of a lipid membrane pore that is characterized by high membrane curvature, accommodating cone-shaped PI(4,5)P₂ molecules. Thus, under these conditions, the opening of the pore reduces membrane stress[31].

Beyond the role of PI(4,5)P₂ transbilayer asymmetry in membrane pore opening as demonstrated by the biochemical reconstitution experiments presented in this study, it is also possible that, in a cellular

context, the addition of PI(4,5)P₂ to the outer plasma membrane leaflet repels the heparan sulfate chains of GPC1, thereby inhibiting FGF2 translocation to cell surfaces. In line with that is our observation that exogenous addition of other phosphoinositides, such as PI(3,4)P₂ and PI(3,4,5)P₃ inhibit FGF2 secretion from cells as well, whereas PI(4)P does not. There is a clear correlation between these phenotypes and the ability of these phosphoinositides to bind to FGF2[6,34]. Thus, when FGF2 membrane translocation is studied in native cells, transbilayer asymmetry of PI(4,5)P₂ may have a dual role with (i) reducing the energetic barrier for membrane pore opening and (ii) ensuring the ability of the heparan sulfate chains of GPC1 to get into close proximity with the outer plasma membrane leaflet, an arrangement that is essential for FGF2 secretion[3,15].

In conclusion, the findings presented in this study provide a compelling model, explaining membrane pore opening by FGF2 oligomer-dependent clustering of PI(4,5)P₂ molecules at the inner plasma membrane leaflet. Our work establishes PI(4,5)P₂ transbilayer asymmetry as a mechanistically defined and physiologically validated regulator of membrane remodeling. It represents one of the rare instances in which lipid asymmetry is shown to exert direct, functional control over a cellular membrane transformation event. Given the central role of PI(4,5)P₂ in coordinating diverse cytosolic processes, its asymmetric distribution may constitute a general and tunable biophysical principle by which cells lower the energetic cost of membrane remodeling events.

## Methods

All the materials used in the study are listed in supplementary files as Supplementary Tables 1 and 2.

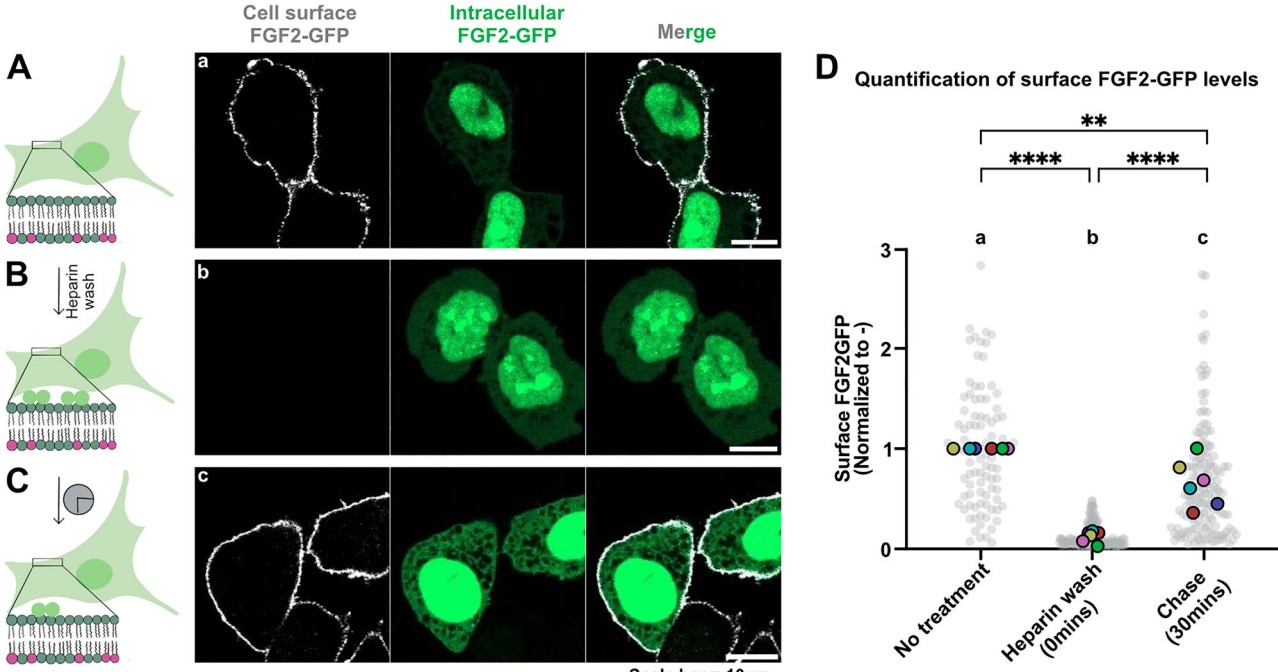

**Fig. 7 | A quantitative assay measuring an acute wave of FGF2 secretion from cells.** Schematic representation of assay used to measure FGF2 secretion from CHO-K1 cells. CHO-K1 cells were induced to express FGF2-GFP overnight by 1 μg/mL doxycycline. **A** FGF2-GFP localization in equilibrium, with FGF2-GFP distributed in the nucleus, cytosol, inner and outer leaflet of plasma membrane. On the right side there is a representative confocal image of fixed cells after overnight doxycycline induction. Surface FGF2-GFP was detected using rabbit polyclonal anti-GFP-Alexa647 antibody at 1:400 dilution. **B** Surface FGF2-GFP was removed using 1 mg/mL heparin. On the right side there is a representative confocal image of fixed cells at 0 min after heparin wash. **C** Recovery of FGF2-GFP on the cell surface was monitored over time post heparin wash. On the right side there is a representative

confocal image of fixed cells at 30 min after heparin wash. Cells were fixed using 3% PFA in PBS for 13 min at room temperature. Using this assay, surface FGF2-GFP can be monitored at the single cell resolution. Left-side cartoons adapted from ref. 56. **D** Quantification and statistical analysis of surface FGF2-GFP levels for individual cells that were fixed without any treatment, fixed 0 min after the heparin wash or 30 min after the heparin wash. Each light grey dot represents a single cell. Mean values from 6 replicates are presented in the solid colors. In each independent experiment, surface FGF2 signal is normalised with the mean of untreated cells. For statistical test, One-way ANOVA was performed on mean values from 6 replicas in Prism. Not significant (ns) $P > 0.5$, ** $P < 0.01$, **** $P < 0.0001$. Data distribution was assumed to be normal. Source data are provided as a Source Data file.

## Formation of Large Unilamellar Vesicles (LUVs)

| Lipid Composition (mol%) |
|---|
| PC + 2% or 3% or 4% or 5%PI(4,5)$P_2$ + 1%RhPE* |
| PC + 5%PI(4)P + 1%RhPE* |
| PC + 10%PE + 5%PS + 5%PI + 30%CHO + 15%SM + 5%PI(4,5)$P_2$ + 1%RhPE* |
| PC + 10%PE + 5%PS + 5%PI + 30%CHO + 15%SM + 5%PI(4)P + 1%RhPE* |

*Liss 16:0 RhPE is used as a fluorescent membrane marker.

All the lipid were purchased from Avanti and stored in chloroform sealed with Argon at −20 °C. Desired lipid concentrations were added to a 25 mL round-bottom flask at final concentration of 4 mM. Chloroform was evaporated under constant nitrogen flow while continuously shaking the flask to generate a thin lipid film. The lipid film was then hydrated using a buffer solution containing 10% (w/v) sucrose, 25 mM HEPES-KOH, 150 mM KCl pH 7.4, followed by vortexing to fully solubilize and hydrate the dried lipid film. To generate unilamellar vesicles, the solution was subjected to 10 freeze-thaw cycles, with each freeze (in liquid $N_2$) and thaw (55 °C water bath) step lasting 5 min. In the final step, the solution was subjected to 27 cycles of extrusion with 400 nm filters. Liposomal size was measured by dynamic light scattering (observed size = 200–400 nm). The resulting

vesicles were aliquoted and stored at −80 °C for long-term storage. Each liposome aliquot was never freeze-thawed more than twice after its initial preparation.

## Formation of Giant Unilamellar Vesicles (GUVs)

| Lipid Composition (mol%) |
|---|
| PC + 5%PI(4,5)$P_2$ + 1%Biotynl-PE** + 0.05%RhPE* |
| PC + 5%PI(4)P + 1%Biotynl-PE** + 0.05%RhPE* |
| PC + 10%PE + 5%PS + 5%PI + 30%CHO + 15%SM + 5%PI(4,5)$P_2$ + 1%Biotynl-PE** + 0.05%RhPE* |
| PC + 10%PE + 5%PS + 5%PI + 30%CHO + 15%SM + 5%PI(4)P + 1%Biotynl-PE** + 0.05%RhPE* |

*Liss 16:0 RhPE is used as a fluorescent membrane marker
**Biotnyl-PE is used for immobilizing GUVs on Biotin-NeutrAvidin coated Ibidi chambers.

The formation of GUVs was adapted from ref. 12 with minor modifications. Lipids at the desired concentration were mixed to prepare a working lipid mix solution. A total of 5 μL of this lipid mix was deposited on platinum electrodes (2.5 μL per electrode; Goodfellow Cambridge Limited 0.5 mm diameter, 99.99% pure, degree of hardening: glowed) (Teflon Chambers were custom-made by MPI Dresden/University Heidelberg). The electrodes were then dried for 10–15 min

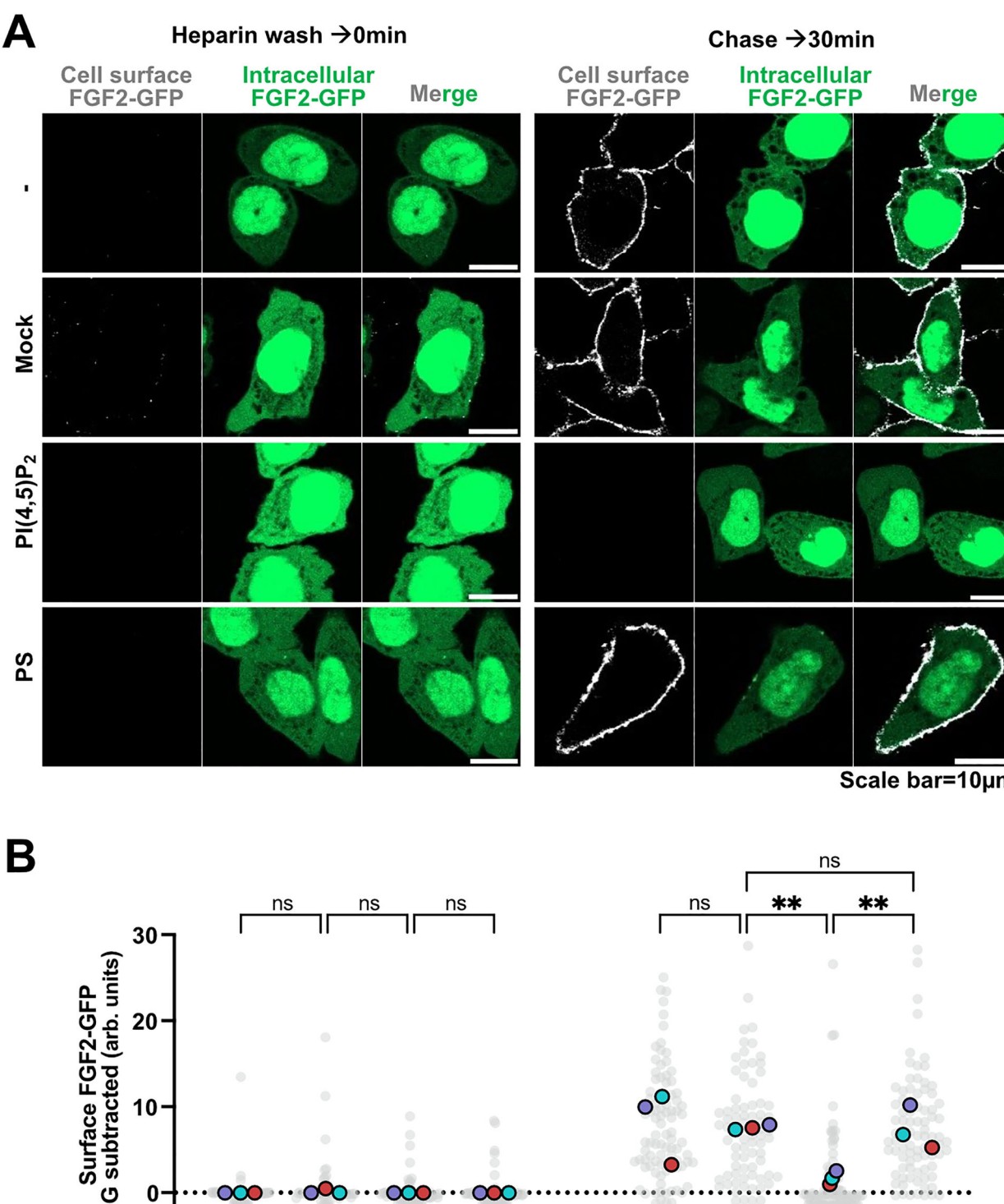

**B**

Scale bar=10μm

before submerging them in a 300 mM sucrose solution (301 Osmol). GUV formation was carried out for 50 min at 1.5 V, 10 Hz, 55 °C. Following formation, GUVs were detached from the electrode surface for 25 min at 1.5 V, 2 Hz, 55 °C. To prevent sudden temperature changes, before washing, the GUVs were gradually cooled to room temperature over 20 min. GUVs were then washed three times with 1200 μL of 25 mM HEPES-KOH, 150 mM KCl pH 7.4 buffer, 306 Osmol (HK buffer)

by centrifugation at 1200 × g for 5 min at 25 °C. After each wash, 1 mL of the supernatant was gently removed from the top, ensuring the lucid pellet remained undisturbed. In the final step, the pellet was resuspended and gently transferred to a glass bottom 8 well Ibidi chambers (80807).

The Ibidi chambers were pre-blocked with 0.1 mg/mL Biotin-BSA for 15 min, followed by three washes with water, and was later blocked

**Fig. 8 | Disruption of transbilayer PI(4,5)P₂ asymmetry inhibits FGF2-GFP secretion in cells.** CHO-K1 cells were induced to express FGF2-GFP overnight by 1 µg/mL doxycycline. **A** Representative confocal images of non-treated, mock-treated, PI(4,5)P₂-treated, and PS-treated cells. Cells were treated with 1 mg/mL heparin to remove surface FGF2-GFP. On left side of the panel are the representative confocal images of fixed cells at 0 min after the heparin wash. On the right side of the panel, there are confocal images of the fixed cells at 30 min after the heparin wash. Cells were fixed using 3%PFA in PBS for 13 min at room temperature. Surface FGF2-GFP was detected using rabbit polyclonal anti-GFP-Alexa647 antibody at 1:400 dilution. **B** Quantification and statistical analysis of surface FGF2-GFP levels for individual cells that were left untreated, mock-treated, PI(4,5)P₂-treated, and PS-treated. Each light grey dot represents a single cell coming from 3 independent experiments. Mean values from 3 replicates are presented in the solid colors. Unspecific background (BG) from antiGFP-AlexFluor-647 signal after heparin wash for each condition is subtracted. For statistical test, One-way ANOVA was performed on mean values from 3 replicas in Prism. Not significant (ns) $P > 0.5$, $** P < 0.01$. Data distribution was assumed to be normal. Source data are provided as a Source Data file.

with 0.1 mg/mL NeutrAvidin for 15 min, followed by three washes with water and once more with HK buffer. Before adding 200 µL of washed GUVs to Ibidi chambers, 100 µL of HK buffer with Alexa 647 dye and His-FGF2-GFP or His-FGF2-Halo at a final concentration of 200 nM was added. The protein was incubated with GUVs for approximately 1–1.5 h before imaging for bulk quantification or for 15–20 min for kinetic measurement.

Confocal microscopy: Imaging was done with Zeiss LSM-800 using 1.4 oil immersion lens with 63× magnification. Two track-three channel was used for imaging, Channel 1-RhPE (561) (red-membrane marker to identify GUV) Channel 2-FGF2-GFP or FGF2-Halo-Alexa488 (488 nm) (green-FGF2 fusion protein to analyze FGF2 membrane binding) Channel 3-Alexa647 (647 nm) (gray-a small ≈1KDa dye which can enter lumen of the GUV in the event of pore formation). For all the conditions, GUVs with and without FGF2 fusion proteins were analyzed. For recording time-series for FGF2-GFP translocation kinetics, after adding protein to the GUVs, time was marked as 0 min, GUV of interest was identified and imaged till 1.5–2 h. Time-series was recorded with 30 s, 5 min, or 10 min difference in between the individual frames. After image acquisition, the data were processed and analyzed using Fiji either manually or by a macro script. Images were converted from.czi files into.tif and.jpg formats, with the latter used for visualization purposes.

For quantifying % of GUVs with pore, small circle was drawn inside the GUV and to the immediate outside of the GUV. If ratio of Alexa-647 fluorescence inside to the outside was equal to or above 0.6 mean threshold, GUV was declared to have the pore. For each experimental replica condition 19–135 GUVs were analyzed.

## Protein expression

Recombinant FGF2 fusion proteins and PI(4,5)P₂ detector-PHx2-Halo were purified with E.coli strain BL21 Star using expression vector for His-FGF2-GFP, His-FGF2-Halo, and His-PHx2-Halo. Protein expression was done with 2xYT media for overnight. All the proteins were purified on Ni-NTA affinity chromatography. For His-FGF2-GFP and His-FGF2-Halo, heparin chromatography was performed. In some case, additional, Superdex 75 column was also performed or PD-10 desalting column was done for buffer exchange. For PHx2-Halo protein, after initial Ni-NTA affinity chromatography, protein was subjected to HRV-3C protease to remove His tag. This was followed by Ni-NTA affinity chromatography to remove His-Tag. In the final step, Superdex 75 column was also performed to yield the pure protein. All the proteins were buffered with 25 mM HEPES-KOH, 150 mM KCl, pH = 7.4 with 2% glycerol.

## FACS flow cytometry

FACS procedures were adapted from ref. 34. Prior to the experiment, Eppendorf tubes were blocked with fatty acid-free BSA unless stated otherwise. On the following day, BSA was removed and Eppendorf tubes were washed with 25 mM HEPES-KOH,150 mM KCl pH 7.4 buffer (HK buffer). LUVs at a concentration of 1 mM were washed in 500 µL HK buffer followed by incubation with 50 µL of 2 µM His-GFP, 2 µM His-FGF2-GFP, or 5 µM His-Halo, 5 µM His-FGF2-Halo for 1 h, unless stated otherwise. Halo tag was detected using Halo ligand Alexa488. Unbound protein was washed with 500 µL HK buffer by centrifugation

for 10 min at 16000 × g, 25 °C. Supernatant containing unbound protein was removed carefully without disturbing the pellet. In the final step, pellet was resuspended in 500 µL HK buffer unless stated otherwise. For flow cytometry, liposomes were gated based on size− light scattering−and rhodamine-derived fluorescence−incorporated in the liposomal membrane. Minor background signals from liposomes incubated with GFP or Halo +Alexa488 were subtracted from the signals of liposomes incubated with His-FGF2-GFP or His-FGF2-Halo +Alexa488. The resulting signal was then normalized to 5 mol% PI(4,5)P₂ vesicles.

## Sedimentation assay

1 mM LUVs were washed in 100 µL of 25 mM HEPES-KOH, 150 mM KCl pH 7.4 buffer (HK buffer), followed by incubation with 50 µL of 4 µM His-FGF2-GFP or 5 µM His-FGF2-Halo for 1 h or 3 h. The vesicles were then centrifuged at 16,000 × g for 10 min at 25 °C, and the unbound FGF2 was collected as the supernatant. The pellet, containing vesicles bound to the FGF2 fusion protein, was washed twice with 100 µL HK buffer, and the supernatant was discarded. In the final step, the pellet (containing vesicles bound to the protein) were resuspended in 1× reducing SDS-sample buffer. Both the unbound and bound fractions of the FGF2 fusion protein were analyzed on a 1.5 mm, 4–12% Bis-Tris SDS-PAGE in MES buffer. Proteins were stained with Coomassie instant blue. For analysis, the gel was scanned using an EPSON Perfection 4870 photo scanner or a LICOR Odyssey system. For analysis, FGF2 bound to the 5% PI(4,5)P₂ vesicles, was set to 100% and data was normalized accordingly. Uncropped and unprocessed gels are shown in source data files.

## Kinase reaction to generate asymmetric LUVs/GUVs

Both LUVs and GUVs were subjected to kinase reaction to generate PI(4,5)P₂ asymmetric vesicles. In the first step, GUVs or 1 mM LUVs were washed with 25 mM HEPES-KOH, 150 mM KCl pH 7.4 (HK buffer). To the liposome pellet, premixed reaction ingredients 50 mM TRIS-HCl (pH = 7.4), 5 mM MgCl₂, 100 µM ATP, 20 nM PIP5K1C in HK buffer were added (1 mM liposome ≈ 50 µM PI(4)P or PI(4,5)P₂). Osmolarity of luminal and extra-luminal solution do not differ by more than 10−20 Osmol. Reaction was carried out at 25 °C for 15 min with shaking at 600 rpm for LUVs and 350 rpm for GUVs. Liposomes were washed twice, with 50−500 µL HK buffer for LUVs (depending on the assay) and 1 mL HK buffer for GUVs to remove all the unreacted ingredients or damaged vesicles. Following the reaction, vesicles were analyzed with sedimentation assay, flow cytometry, and confocal imaging.

## Kinase reaction to generate asymmetric GUVs for bulk quantification at 0, 40, and 90 min

For bulk quantification, GUVs were subjected to kinase reaction to generate PI(4,5)P₂ asymmetric vesicles with some modifications to improve upon the yield. After formation of GUVs, they were washed with 1200 µL of 25 mM HEPES-KOH, 150 mM KCl pH 7.4, 305 Osmol (HK buffer) at 1200 × g for 5 min at 25 °C. After the centrifugation, supernatant was gently removed from the top, ensuring the lucid pellet (≈450−500 µL) remained undisturbed. In the final step, the pellet was resuspended and 200 µL GUVs were gently transferred to a BiotinBSA/ NeutrAvidin pre-blocked 8 well Ibidi chambers (See formation of GUVs

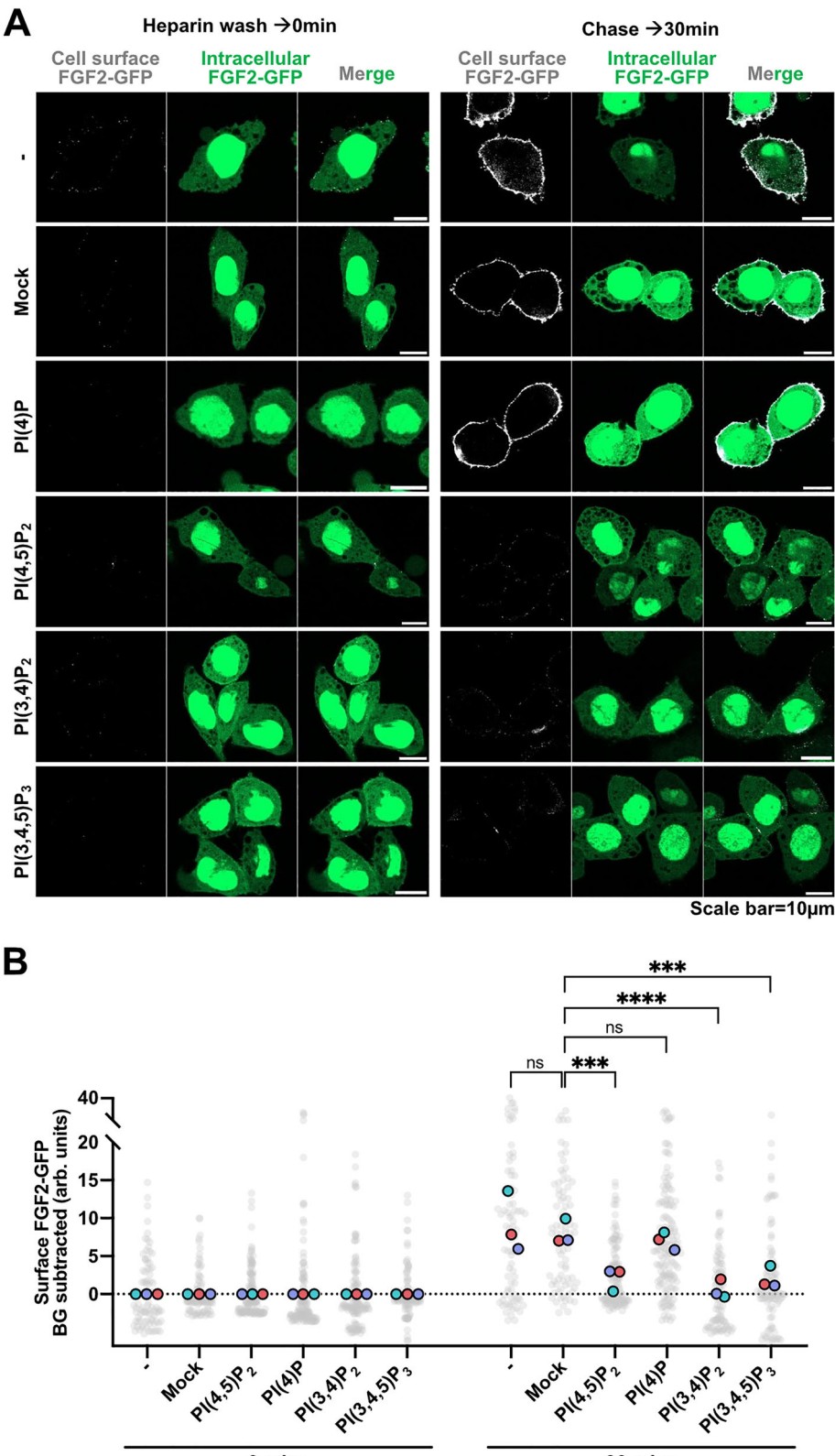

section for more details). After 25 min, most of GUV would have immobilized on the glass surface. For cleaning, 300 µL HK buffer was pipetted to each chamber and the same amount was taken out. This step was repeat 3 time. For the kinase reaction 80 µL of kinase premix was added in total volume 200 µL, so final solution has 50 mM TRIS-HCl (pH = 7.4), 5 mM $MgCl_2$, 100 µM ATP, 20 nM PIP5K1C. Osmolarity of luminal and extra-luminal kinase premix solution do not differ by

more than 5–10 Osmol. Reaction was carried out at 25 °C for 15 min with gentle shaking. For the washing, 300 µL HK buffer was pipetted to each chamber and the same amount was taken out. This step was repeat 5–6 time for effective cleaning. All the chambers were treated through same steps of washing procedure. In the final step, vesicles were treated with Alexa647 dye or 200 nM FGF2-GFP along with Alexa647 dye. To visualize FGF2-GFP dependent pore formation, % of

**Fig. 9 | Effects of exogenously added phosphoinositides on FGF2-GFP secretion from cells.** CHO-K1 cells were induced to express FGF2-GFP overnight by 1 μg/mL doxycycline. **A** Representative confocal images of non-treated, mock-treated, PI(4,5)P$_2$-treated, PI(4)P-treated, PI(3,4)P$_2$-treated and PI(3,4,5)P$_3$-treated cells. Cells were treated with 1 mg/mL heparin to remove surface FGF2-GFP. On left side of the panel are the representative confocal images of fixed cells at 0 min after the heparin wash. On the right side of the panel there are confocal images of the fixed cells at 30 min after the heparin wash. Cells were fixed using 3%PFA in PBS for 13 min at room temperature. Surface FGF2-GFP was detected using rabbit polyclonal anti-GFP-Alexa647 antibody at 1:400 dilution. **B** Quantification and statistical analysis of surface FGF2-GFP levels for individual cells that were left untreated, mock-treated, PI(4,5)P$_2$-treated, PI(4)P-treated, PI(3,4)P$_2$-treated, and PI(3,4,5)P$_3$-treated. Each light grey dot represents a single cell coming from 3 independent experiments. Unspecific background (BG) from antiGFP-AlexFluor-647 signal after heparin wash for each condition is subtracted. Mean values from 3 replicates are presented in the solid colors. For statistical test, One-way ANOVA was performed on mean values from 3 replicas in Prism. Not significant (ns) $P > 0.5$, *** $P < 0.001$, **** $P < 0.0001$. Data distribution was assumed to be normal. Source data are provided as a Source Data file.

vesicles filled with Alexa647 dye were subtracted from GUVs that were treated with 200 nM FGF2-GFP and Alexa647. For GUV imaging/analysis/quantification see Formation of Giant Unilamellar Vesicles (GUVs) section. It is important to note that the imaging was started at 0, 40, and 90 min. To take enough GUVs for bulk quantification for all conditions, each imaging session for 3 time periods took 25–30 min. Representative confocal images demonstrating kinase reaction success are provided in the source data files.

### Testing lipid asymmetry in LUVs

PC + 5%PI(4,5)P$_2$ (positive control for FGF2 binding) and PC + 5%PI(4)P (substrate liposome) vesicles were used as controls. For kinase reaction 50 mM TRIS-HCl, 5 mM MgCl$_2$, 100 μM ATP, 1 mM liposomes, and 20 nM PIP5K1C was used. As additional controls, reaction ingredients were added to the PI(4,5)P$_2$ vesicles and mock was performed with PI(4)P vesicles. After completion of the reaction, reaction mixture was suspended in 50 μL of 25 mM HEPES-KOH, 150 mM KCl pH 7.4 (HK buffer) and centrifuged at 16000 × $g$, 10 min at 25 °C. Supernatant was discarded and the washing step was repeated once more.

For non-treated samples, post washing, the pellet was resuspended in 50 μL of 1 μM FGF2-GFP or 1 μM GFP for 30 min. After incubation, 50 μL of HK buffer was added, and the sample was centrifuged at 16,000 × $g$ for 10 min at 25 °C. The supernatant was discarded, and the washing step was repeated one more time. In the final step, the pellet was resuspended in 350 μL of HK buffer for FACS measurement. (Note: the pellet can be stubborn when binding occurs, so it is important to resuspend it a minimum of 50–100 times.)

For reformation, the washed pelleted vesicles were disintegrated with 50 μL HK buffer and 100 μL of a 40% MeOH/CHCl$_3$ solution using 5 s of vortexing. The organic and aqueous layers were allowed to separate for 2–5 min. The organic layer was carefully collected using a pipette (the tip should be pre-filled with the 40% MeOH/CHCl$_3$ solution to avoid sucking in the aqueous phase). The aqueous layer was washed twice more with 100 μL of 40% MeOH/CHCl$_3$ to ensure complete transfer of lipid content to the organic layer. The organic layer was then dried by purging with nitrogen gas and was placed under vacuum for at least 2 h. The dried lipid film was hydrated with 50 μL of 10% (w/v) sucrose HK buffer. To generate unilamellar vesicles, sample was subjected to 10 freeze-thaw cycles. After this stage, 5 μL of the sample was collected for dynamic light scattering measurement. The sample was then diluted with 100 μL of HK buffer and centrifuged at 16,000 × $g$ for 10 min at 25 °C. After centrifugation, the pellet was resuspended in 50 μL of 1 μM FGF2-GFP or 1 μM GFP for 30 min. After incubation, 50 μL of HK buffer was added, and the sample was centrifuged at 16,000 × $g$ for 10 min at 25 °C. The supernatant was discarded, and the washing step was repeated one more time. In the final step, the pellet was resuspended in 350 μL of HK buffer for FACS measurement. FACS flow cytometry measurements were performed for all liposome samples containing GFP or FGF2-GFP. For analysis, FGF2-GFP bound to the 5% PI(4,5)P$_2$ vesicles, was set to 100% and data was normalized accordingly.

### Testing lipid asymmetry in GUVs

GUVs with PM + 5 mol% PI(4,5)P$_2$ and PM + 5 mol% PI(4)P were prepared, washed (once), and immobilized on glass bottom ibidi

chambers as described previously in this methods section. Vesicle was identified using Zeiss LSM-800 using 1.4 oil immersion lens with 63× magnification. Imaging involved two channels: Channel 1-RhPE (561) (red-membrane marker to identify GUV) Channel 2-Alexa647 (647 nm) (gray-a small ≈1KDa dye which can enter lumen of the GUV in the event of pore formation). To the GUVs 5 mM MgCl$_2$, 100 μM ATP, 20 nM PIP5K1C (see "*Kinase reaction to generate asymmetric LUVs/GUVs*" section) along with tracer dye Alexa647 was added and time was marked as 0 min. Vesicle was monitored for 15 min and time series was recorded with 1 min difference in each individual frame.

### Culturing cells CHO-K1

CHO-K1 cells, expressing FGF2-GFP in a doxycycline-dependent manner were used. The cells were cultured in Minimal Essential Medium Eagle (α-MEM) supplemented with 10% heat inactivated Fetal Calf Serum (FCS), 2 mM glutamine, 100 U/mL penicillin, 100 μg/mL streptomycin. Cells were grown at 37 °C with 95% humidity and in the presence of 5% CO$_2$. They were maintained for 20 passages or approximately two months after thawing. Before the experiment, FGF2-GFP expression was induced with 1 μg/mL doxycycline for overnight. For CHO-K1-FGF2GFP cells, identity and purity were analyzed by a multiplex cell contamination test.

### Disrupting transbilayer PI(4,5)P$_2$ and PS asymmetry in cells

Approximately, 10000 cells were seeded in a glass bottom 8 well ibidi chambers (80807) with 1 μg/mL doxycycline, overnight prior to experiment. Lipids at the desired concentration were placed in an Eppendorf tube and dried for approximately 1 h under vacuum. They were then resuspended in a 6 mM EDTA solution and processed with the following cycle: 30 s of vortexing, 5 min in a 55 °C water bath, 30 s of vortexing, and 5 min of sonication. This cycle was repeated once more. Micelle/vesicle size was measured using dynamic light scattering (observed size = multi-populational distribution below and above 100 nm).

PI(4,5)P$_2$ treatment: Cells were treated with 15 nM lipid in 1.5 mM EDTA using FCS-free media, with a final volume of 50 μL lipid EDTA solution in 150 μL FCS-free media per chamber. The cells were incubated at 37 °C for 20 min.

Mock treatment: Cells were treated with 1.5 mM EDTA using FCS-free media, with a final volume of 50 μL of EDTA solution in 150 μL FCS-free media per chamber. The cells were incubated at 37 °C for 20 min.

No treatment: Cells were not treated with any solution.

[Calculation description: For PI(4,5)P$_2$−Total lipids in PM assumed to be $10^9$ lipids, assuming 3% PI(4,5)P$_2$ in the PM, three times more PI(4,5)P$_2$ was added in the experiment. EDTA was added because of Ca$^{2+}$ and Mg$^{2+}$ in the media. Divalent cations aggregate PI(4,5)P$_2$, which hinders the incorporation of PI(4,5)P$_2$ in the outer leaflet. Total Ca$^{2+}$+ Mg$^{2+}$ in media is approximately ≈ 1.5 mM, so 1.5 mM EDTA was used in the experiment.

For PS lipid: It is used as a control lipid, and was added similar to PI(4,5)P$_2$.

For phosphatidylinositol derivatives: PI(4)P, PI(3,4)P$_2$, PI(3,4,5)P$_3$, similar concentrations to that of PI(4,5)P$_2$ were added]

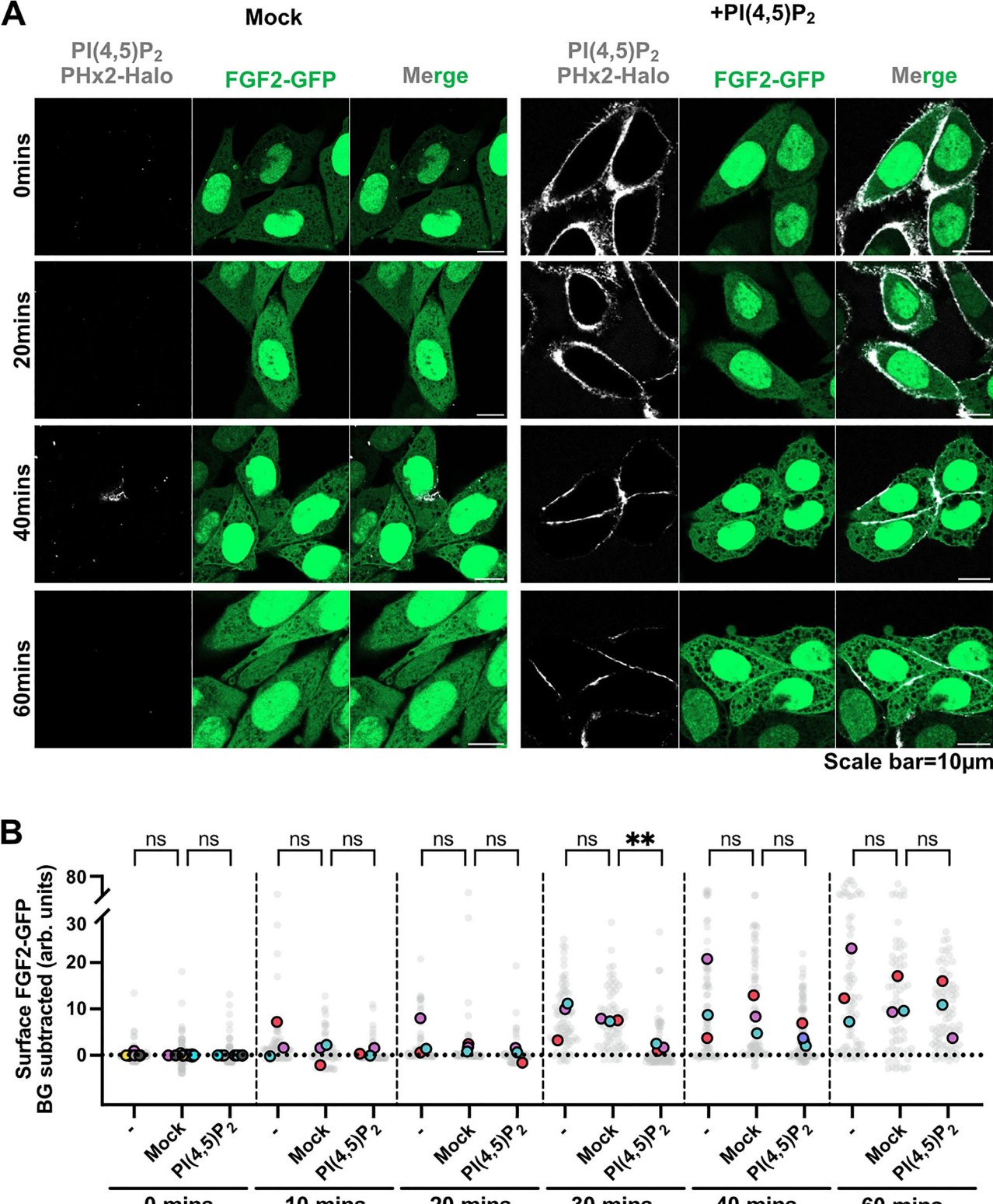

**Fig. 10 | FGF2-GFP secretion is modulated by PI(4,5)P₂ levels at the plasma membrane surface.** CHO-K1 cells were induced to express FGF2-GFP overnight by 1 μg/mL doxycycline. **A** Representative confocal images of mock-treated and PI(4,5)P₂-treated CHO-K1 cells. Cells were fixed either at 0, 20, 40, and 60 min after disruption of plasma membrane PI(4,5)P₂ asymmetry. Fixed cells were stained with high affinity PI(4,5)P₂ binding protein PHX2-Halo (40 nM). The Halo tag was visualized using Halo ligand Alexa 660 (75 nM). **B** Quantification and statistical analysis of surface FGF2-GFP levels for individual cells that were left untreated, mock-treated, and PI(4,5)P₂-treated. Cells were fixed at 0, 20, 30, 40, and 60 min after the heparin wash. Cells were fixed using 3%PFA in PBS for 13 min at room temperature. Surface FGF2-GFP was detected using rabbit polyclonal anti-GFP-Alexa647 antibody at 1:400 dilution. Each light grey dot represents a single cell coming from 3 to 9 independent experiments. Mean values from different replicates are presented in the solid colors. Unspecific background (BG) from antiGFP-AlexaFluor-647 signal after heparin wash for each condition was subtracted. For statistical test, One-way ANOVA was performed on mean values from 3 replicas in Prism. Not significant (ns) $P > 0.5$, ** $P < 0.01$. Data distribution was assumed to be normal. Source data are provided as a Source Data file.

After each treatment, cells were washed with PBS three times. After this, either cells were fixed with 3%PFA for 13 min at r.t. or were kept live in LCI solution.

For detecting $PI(4,5)P_2$, 40 nm PHx2-Halo –detected with Alexa fluor 660– was used for 25–30 min at r.t. Before imaging, cells were washed three times with PBS to remove unbound protein and were kept in PBS (for fixed cell imaging) or LCI (for live cell imaging). For PS detection, only live cells could be used, as fixing of cells led to PS exposure on the exoplasmic leaflet. AnnexinV-Alexa647 with 2 mM $Ca^{2+}$ in LCI solution was used as PS detector protein at 1.5:100 dilution. Imaging was done instantly after addition of Annexin V to cells. $PI(4)P$, $PI(3,4)P_2$, $PI(3,4,5)P_3$ could also be detected with high affinity PHx2-Halo (40 nM) –detected with Alexa fluor 660. Cells were incubated for 30 min at r.t. before washing with PBS three times. Representative confocal images demonstrating successful incorporation of $PI(4)P$, $PI(3,4)P_2$, $PI(4,5)P_2$, $PI(3,4,5)P_3$ are provided in the source data files.

For time course study, the goal was to observe retention of PS and $PI(4,5)P_2$ on the outer leaflet of PM over time. After disturbing the $PI(4,5)P_2$ lipid, cells were fixed at either 0, 20, 30, 40, 60 min and analyzed with PHx2-HaloAlexa fluor 660.

Imaging and processing: Cells were imaged using Zeiss LSM-800 using 1.4 oil immersion lens with 63× magnification. Two track channel was used for imaging, Channel 1-PHx2-Halo detection with Alexa660 (660) or AnnexinV-Alexa647 (647 nm) (white) and Channel 2-FGF2-GFP (488 nm) (green). After image acquisition, the data were processed and analyzed using Fiji. A macro script was used to convert the.czi files from the LSM800 microscope into.tif and.jpg formats, with the latter used for visualization purposes.

### FGF2 Translocation assay using confocal microscope

Approximately 10000 cells were seeded with 1 μg/mL doxycycline, overnight prior to the experiment. On the following day, cells were washed with PBS three time. Cells were either left untreated, mock-treated, or treated with $PI(4,5)P_2$ PS, $PI(4)P$, $PI(3,4)P_2$, and $PI(3,4,5)P_3$ as described above. Following the treatment, cells were washed three times with PBS, then incubated with 1 mg/mL heparin for 10 min at room temperature. For effective heparin wash, indicated cell density was crucial. After another three PBS washes, cells were placed in LCI for 0 or 30 min, followed by fixation with 3% PFA for 13 min at room temperature. To visualize surface FGF2-GFP, cells were incubated with an anti-GFP-Alexa647 antibody at 1:400 dilution in PBS for 30 min at room temperature. This was followed by three washes with PBS, after which the cells were stored in PBS and imaged using confocal microscopy.

For more elaborate time course FGF2 translocation at different time points, cells were either left untreated, mock-treated, or treated with $PI(4,5)P_2$ as described above. Following the treatment, cells were washed three times with PBS, then incubated with 1 mg/mL heparin for 10 min at room temperature. After another three PBS washes, cells were placed in LCI for 0, 10, 20, 30, 40, or 60 min, followed by fixation with 3% PFA for 13 min at room temperature. To visualize surface FGF2-GFP, cells were incubated with an anti-GFP-Alexa647 antibody at 1:400 dilution in PBS for 30 min at room temperature. This was followed by three washes with PBS, after which the cells were stored in PBS and imaged using confocal microscopy.

Imaging and quantification: Cells were imaged using Zeiss LSM-800 using 1.4 oil immersion lens with 63× magnification. Two track channel was used for imaging, Channel 1-AntiGFP-Alexa647 (647 nm) (white) and Channel 2-FGF2-GFP (488 nm) (green). For each condition, 15–100 cells were imaged.

After image acquisition, the data were processed and analyzed using Fiji. A macro script was used to convert the.czi files from the LSM800 microscope into.tif and.jpg formats, with the latter used for visualization purposes. For quantifying surface FGF2-GFP–detected via antiGFP-Alexa647–the macro script was applied to the.tif files. After

setting an appropriate threshold, individual cells in each image were identified and saved as Regions of Interest (ROIs). A band was drawn near the cell boundary to account for the cellular membrane, and the intensity of anti-GFP-Alexa647 was recorded. Each ROI was manually inspected to ensure accurate cell selection. In cases of misidentification, the ROI was adjusted and updated accordingly. As a final step, the mean from "surface FGF2-GFP after heparin wash at 0 min" condition was subtracted from cells at all time points to account for nonspecific background unless stated otherwise. The processed data were plotted in Prism. Statistical analysis was performed on mean values obtained from each replica using an ordinary one-way ANOVA test.

### Statistical analysis

For statistical test, ordinary one-way ANOVA was performed in GraphPad Prism. Not significant (ns) $P > 0.5$, *$P < 0.05$, **$P < 0.01$, ***$P < 0.001$, ****$P < 0.0001$. Data distribution was assumed to be normal.

### Reporting summary

Further information on research design is available in the Nature Portfolio Reporting Summary linked to this article.

## Data availability

Uncropped gel images and original video files are included in the Source Data files. Details of the corresponding data files are specified in each figure legend. Source data are provided with this paper.

## Code availability

The Fiji macros used in this study have been deposited in the Zenodo repository [https://doi.org/10.5281/zenodo.17306345]. The Fiji macros are publicly available without restriction[55].

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

## Acknowledgements

We thank André Nadler (MPI-CBG Dresden, Germany) and Pavel Barahtjan, providing an expression construct encoding a fusion protein of a PH domain and the Halo tag as well as Hans-Michael Müller for a high-quality preparation of this fusion protein. This work was supported by the Deutsche Forschungsgemeinschaft (SFB/TRR 186, project A1; W.N. and F.L., SFB-1638/1 – 511488495 - P11; W.N., DFG Ni 423/10-1, DFG Ni 423/12-1, and DFG Ni 423/13-1; W.N. and SFB-1638/1 – 511488495 - Z01; F.L., DFG LO 2821/1-1; F.L.). F.L. and W.N. gratefully acknowledge the data storage service SDS@hd supported by the Ministry of Science, Research, and the Arts Baden-Württemberg (MWK), the German Research Foundation (DFG) through grant INST 35/1314-1 FUGG and INST 35/1503-1 FUGG. For the publication fee we acknowledge financial support by Heidelberg University.

## Author contributions

M.K. contributed to experimental design, performed all experiments and data analysis as well as contributed to writing the manuscript. F.L. and W.N. developed the scientific concept, designed and supervised experiments, secured funding, and wrote the manuscript. All authors discussed data analysis, statistical evaluation and data presentation as well as gave final approval for publication.

## Funding

## Competing interests

The authors declare no competing interests.
