## [Transparent Peer Review file · Nature Communications]

Plasma membrane transbilayer asymmetry of PI(4,5)P₂ drives unconventional secretion of Fibroblast Growth Factor 2

Corresponding Author: Dr Fabio Lolicato

Version 0:

Reviewer comments:

Reviewer #1

(Remarks to the Author)

This manuscript addresses the mechanism of unconventional protein secretion, focusing on FGF2 translocation across the plasma membrane—a topic of increasing interest, albeit still in its early stages. The authors, with a longstanding commitment to this field, have previously demonstrated the role of PI(4,5)P₂ in regulating FGF2 membrane translocation. In the current study, they significantly extend their model by demonstrating that asymmetric distribution of PI(4,5)P₂ across the lipid bilayer is critical for FGF2-induced pore formation and subsequent translocation.

They further developed technically elegant approaches to generate bilayer asymmetry both in vitro and in cells, and established a novel pulse-labeling assay for monitoring FGF2 secretion dynamics in live cells. The study is highly original and methodologically innovative, with findings that contribute significantly to our understanding of unconventional secretion. I believe the manuscript is suitable for publication in Nature Communications after addressing the following points:

Use of GUV System: A previously established GUV-based FGF2 translocation assay is not included. It is recommended to directly test the effect of asymmetric PI(4,5)P₂ using the GUV platform to support the in vitro findings.

Figure 1B, C – Discrepancy Between FACS and Western Blot: The FACS-based binding assay in Figure 1B shows a notably different fold change in the middle experimental group compared to the western blot in Figure 1C. Please clarify the cause of this discrepancy.

Statistical Analysis: Statistical significance should be reported for all bar graphs, including Figures 1B/C, 2B/C, and 3A. This will strengthen the conclusions drawn from these data.

Quantification of GUV Fluorescence: Quantitative analysis of GUV fluorescence is recommended for Figures 1D/E and 2B–E to substantiate the visual interpretations.

Figure 2B – Color Labeling: Please label the gray and pink traces/bars in Figure 2B for clarity.

Figure 2B – Interpretation of Group D: In group D of Figure 2B, the gray and pink signals appear nearly identical, although the reassembled (pink) group is expected to show a ~2-fold decrease. Please clarify or reconcile this apparent inconsistency.

Magnesium Ion Effects: In Figure 2B, group C shows enhanced FGF2 binding upon Mg²⁺ removal, which is interpreted as increased PI(4,5)P₂ availability. However, in Figures 1C and 3A, the presence of Mg²⁺ in the middle group appears to reduce FGF2 binding relative to the control. Please explain this seeming contradiction.

Figure 4A/B – Patches of Fluorescence: The presence of fluorescent patches in Figure 4A/B may suggest lipid phase separation. The authors should comment on whether phase separation is occurring and how it might impact their interpretation.

GUV Leakage Quantification: In Figure 4B, it is suggested to perform an additional experiment using a common time point (e.g., 40 minutes) to quantify the percentage of GUV leakage under different treatments.

Lipid Specificity Controls: In Figure 5 and related panels, PS is used as a control lipid. A more appropriate control would be PI(3,5)P₂, which is structurally similar to PI(4,5)P₂. This would more convincingly demonstrate the specificity of FGF2 for PI(4,5)P₂.

Reviewer #2

(Remarks to the Author)

The manuscript by Kaur and Nickel combined an elegant in vitro membrane pore forming assay and a cell-based secretion assay to study the mechanism by which the soluble protein FGF2 destabilizes the membranes. FGF2 was one of the cytosolic proteins known to undergo unconventional secretion by translocation across the plasma membrane (PM) directly. Previous studies from the same group have suggested that FGF2 is recruited to the PM by binding to PI(4,5)P₂. Upon membrane binding, FGF2 moves across the membrane via a pore formed by FGF2 oligomerization. The precise pore forming mechanism by FGF2 is unclear. The authors tested the hypothesis that PI(4,5)P₂ asymmetry in the PM destabilizes the membrane, which allows the FGF2 oligomer to generate pores in the PM. The in vitro cell free assay used PIP5 kinase to convert PI4P in the outer leaflet of the GUV or LUV to PI(4,5)P₂. This generated GUV or LUV with PI(4,5)P₂ only in the outer leaflet, which was demonstrated by FGF2 binding. They showed that the presence of PIP5 kinase together with ATP and Mg⁺⁺ enhanced the pore forming efficiency of FGF2. They also developed a new cell-based assay to monitor the kinetics of FGF2 secretion following heparin wash and chase. They showed that feeding cells with PI(4,5)P₂ micelles could allow the outer leaflet of the PM to gain PI(4,5)P₂, which disrupts its asymmetry. Under this condition, FGF2 secretion is significantly inhibited.

The study design is novel and the data are convincing. The findings significantly improve our understanding of how lipid composition of the membrane can affect protein translocation in unconventional protein secretion. There are a few minor points that the authors should address as listed below.

Main points:

1. Although the in vitro study is elegant and the results are convincing, I feel that a few negative controls should be added to exclude the possibility that the presence of PIP5 kinase, ATP and Mg, collectively enhances the pore forming efficiency independent of lipid asymmetry. The authors used a condition that has PI(4,5)P₂, kinase, ATP and Mg, which showed no effect of the kinase on pore forming. However, the kinase presumably would not bind PI(4,5)P₂, but in the reaction that contained PI4P, the kinase should bind GUV due to the presence of its substrate. This may somehow affect FGF2 binding and pore formation independent of its kinase activity. This possibility should be excluded with additional controls such as a catalytically inactive kinase mutant or a kinase inhibitor.
2. In Figure 2B, d group, why didn't the authors observe reduced FGF2 binding after vesicle reformation? Presumably, the PI(4,5)P₂ concentration in the outer leaflet should be reduced by 50% after vesicle reformation. Oddly, they claimed in line 168 that "The efficiency of FGF2 binding to LUVs went down substantially", but the figure did not show that. Can they repeat this experiment with the gel-based assay to avoid the confounding effect of Mg⁺⁺?
3. For Figure 4D, to better reveal the difference in kinetics, the authors should measure the percentage of permeabilized GUV at an early time point.

Minor points:

1. Line 82, the authors should state the physiological concentration of PI(4,5)P₂ in the PM and justify the concentration used in the in vitro study in the method part.
 2. Line 124, when LUV and GUV were mentioned, they need to define these acronyms. They sometimes used LUV and other times used GUV, but the rationale of using one vs. the other was not explained.
 3. Lines 178-207, this section is confusing. Again, they used both LUV and GUV, but did not explain why. In line 187, they mentioned that they switched to using Halo-tagged FGF2, but Figure 3B, C showed the binding of GFP-tagged FGF2.
 4. They should revise this part to improve the clarity.
- They should provide statistical data for the bar graphs in Figure 1-3 when a comparison was mentioned.

Reviewer #3

(Remarks to the Author)

The manuscript by Kaur et al titled "Plasma membrane transbilayer asymmetry of PI(4,5)P₂ drives unconventional secretion of Fibroblast Growth Factor 2" seeks to further understand the mechanistic process of unconventional secretion of FGF2. The corresponding author's lab has many publications showing that FGF2 is secreted in a manner differently from other cytokines in that it is not mediated by vesicular transport. Instead the growth factor is cytosolic, binds to the inner leaflet of the plasma membrane (PM), and translocates to the exoplasmic leaflet where it binds to heparin-containing glycoproteins. In this manuscript, the authors seek to test whether the membrane translocation of the growth factor is due to the asymmetric distribution of PI(4,5)P₂, a lipid central in FGF2-PM binding that is only located on the cytosolic leaflet of the plasma membrane. The authors use several synthetic systems to show FGF2-PI(4,5)P₂ binding and subsequent pore formation is dependent on an asymmetric distribution of PI(4,5)P₂. They then test this hypothesis in cellular systems. The data is clean and clearly presented, but I do feel a few controls are missing. Additionally, some methodology is missing from the Materials and Methods section making it difficult to assess the results.

Major Comments:

1. In Figure 1, the authors show two methods of quantifying FGF2 binding. In the flow cytometry method, they see a clear experimental hurdle in that the liposomes cluster in Mg²⁺-containing buffers (such as used for the enzymatic studies) which gives an artificially lower FGF2-binding readout. This intensity is vital for quantifying whether FGF2 can bind in a PIP₂-asymmetric manner, and the off-hand mention that the data is lower due to clustering/clumping is insufficient. The authors should show the flow cytometry data that suggests this is the case. To alleviate this, can the authors do a buffer-exchange protocol to remove Mg²⁺ from the buffer after the enzymatic reaction? Additionally, given this potentially artifactual result, why did the authors use this protocol in Fig 2, but used the sedimentation protocol in Fig 3? It would be nice to be able to compare across these tests, and this could be done by using both experimental techniques in Fig 2 and 3.
2. It appears that the vesicles in Fig 3 are uniform while the vesicles in Fig 4 are phase-separated. Is this due to time spent incubating with FGF2, or is this specific to this particular experiment? Can the authors expand on this?
3. The authors spend several figures to show that asymmetric distribution of PI(4,5)P₂ is essential for FGF2 binding in reconstituted systems and that FGF2 won't bind to PI4P-containing vesicles. They then switch to using PS as a control in cell experiments. Can the authors add extracellular PI4P to cells (and test via a PI4P-binding protein in Fig 5) to show that it is indeed PI(4,5)P₂-driven?
4. In general, the cell images shown could benefit from more quantification. Can the authors quantify the images in Fig 6 to show the level of binding, release by heparin, and re-externalization over time?
5. The statistics in Fig 7 and 8 are either confusing or misleading. The figure caption suggests that the dots are individual cells, with data from multiple experiments combined into one graph and statistical test: "each dot represents a single cell coming from 3 independent experiments". This is inappropriate and rather each experiment should be shown independently (e.g. each data point can be a different symbol color or shape). Importantly, the means of the experiments should be shown. And the statistics should be done only on the means of the experiments, and not all of the cells from the combined experiments.
6. It seems that in the video of the vesicle that became leaky to Alexa-647, blebs are pinching off in both directions. Can the authors briefly comment on this in the discussion? Is this essential for FGF2 translocation?

Minor Comments:

1. The legend for the color of the bars in Fig 2B is missing and only stated in the caption. It would be nice to have this information on the graph itself.
2. The concentration of doxycycline should be stated in the methods, not "1:1000 doxycycline".

Version 1:

Reviewer comments:

Reviewer #1

(Remarks to the Author)

Accept

Reviewer #2

(Remarks to the Author)

The authors have addressed all my criticisms. I support the publication of this paper in Nature Communications.

Reviewer #3

(Remarks to the Author)

The authors have addressed all of my concerns.

Reviewer #1

1.) *GUV-based FGF2 translocation assay to study the role of PI(4,5)P₂ transbilayer asymmetry in FGF2 membrane translocation*

We indeed originally planned to study a possible functional role of PI(4,5)P₂ transbilayer asymmetry not only in the context of FGF2-induced membrane pore formation but also using FGF2 membrane translocation as a read-out, an experimental system we published previously (Steringer et al 2017, eLife). Unfortunately, we experienced technical problems when trying to combine luminal inclusion of heparin molecules into PI(4)P-containing GUVs with subsequent PI5K1C-mediated conversion of PI(4)P into PI(4,5)P₂. It is currently unclear as to why the kinase reaction is inefficient under these experimental conditions. Nevertheless, in GUVs with a symmetric transbilayer distribution of PI(4,5)P₂, the kinetics of FGF2-induced membrane pore formation and the kinetics of FGF2 membrane translocation can be recorded simultaneously. As shown in the figure below for two representative examples, membrane pore formation and FGF2 membrane translocation are kinetically tightly coupled with membrane pore formation typically preceding FGF2 membrane translocation by just about 1-2 minutes. Therefore, we are confident that the kinetics of membrane pore formation are an accurate reflection of the subsequent step of FGF2 membrane translocation that occurs almost instantly after membrane pore formation happened. We therefore think that the data presented in our manuscript are fully conclusive. Nevertheless, as suggested by this Reviewer, in future studies we will attempt to master the technical challenges we encountered, aiming at simultaneous measurements of membrane pore formation and FGF2 membrane translocation also for GUVs with an asymmetric PI(4,5)P₂ transbilayer distribution.

2.) *Discrepancies between FACS- and sedimentation analyses in experiments shown in Fig. 1B and 1C*

It is true that, in the presence of PIP5K1C, ATP and Mg²⁺, there are differences in apparent signal strength for FGF2-GFP binding between the FACS and the sedimentation experiments analysed by Coomassie-stained SDS gel electrophoresis that are shown in Fig. 1B and 1C. This is because the Mg²⁺ ions induce clustering of LUVs, having an impact on the absolute signals in the FACS readout. However, importantly, for both the FACS and the Coomassie-stained SDS gel analysis, the signals for symmetric LUVs with PI(4,5)P₂ containing all components of the kinase reaction (green bars) and those for asymmetric LUVs in which PI(4)P was enzymatically converted into PI(4,5)P₂ (violet bars) are similar in terms of FGF2-GFP binding. This demonstrates that PI(4)P can be enzymatically converted into PI(4,5)P₂, producing asymmetric LUVs that have similar amounts of PI(4,5)P₂ on their surfaces compared to symmetric LUVs with PI(4,5)P₂. These findings are consistent with the GUV analyses using confocal microscopy shown in Fig. 1D and 1E, demonstrating similar FGF2-GFP binding efficiencies between asymmetric GUVs produced

by enzymatic conversion of PI(4)P into PI(4,5)P₂ and symmetric GUVs in which PI(4,5)P₂ was included during their formation.

3.) *Analysis of statistical significance in data sets shown in Fig. 1B, 1C, 2B, 2C and 3A*

As proposed by Reviewer 1, we have **updated Figs. 1, 2 and 3** providing a statistical analysis including p-values to test whether differences observed for key conditions are statistically significant.

4.) *Quantification of fluorescence associated with GUVs shown in Figs. 1D, 1E and 3B to 3E*

As requested by Reviewer 1, with the **updated Figs. 1D, 1E and 3B to 3E** the revised manuscript now contains quantitative data on fluorescence intensities associated with GUVs in the figures mentioned above.

5.) *Labelling of bar graphs shown in Fig. 2B*

We apologize for the inconvenience. For some reason, the colour code within Fig. 2B got lost during the preparation of the final version of the original manuscript. This has been corrected in the revised version, making this figure now fully self-explanatory.

6.) *Intensity changes in FGF2-GFP binding to LUVs containing PI(4)P and treated with PIP5K1C, ATP and Mg²⁺ in group d of Fig. 2B*

The signals from the FACS read-out shown in grey bars (before reformation of LUVs) and light red bars (after reformation) are not directly comparable as Mg²⁺ ions (that affect the absolute FACS signals for FGF2-GFP binding) are removed during the liposome reformation procedure. The relevant comparison is therefore the FGF2-GFP binding signal for the light red bars in group d (asymmetric PI(4,5)P₂) with the light red bars in group a and c (symmetric PI(4,5)P₂). Here, for the light red bars, the signal drops substantially from group a/c to group d since the reformation procedure causes a dilution of PI(4,5)P₂ only for asymmetric PI(4,5)P₂ LUVs. To further analyse this phenomenon, we generated a standard curve of FGF2-GFP binding to LUVs as a function of the mol percentage of PI(4,5)P₂ (Fig. 2C). The signal obtained for reformed LUVs from group d (light red bars; Fig. 2B) lies between 2 and 3 mol% PI(4,5)P₂ of the standard curve shown in Fig. 2C. This is in line with a surface reduction of PI(4,5)P₂ from about 5 mol% before reformation to about 2.5 mol% after reformation. These data are consistent with the asymmetric nature of LUVs in which 5 mol% PI(4)P molecules were added during formation followed by enzymatic conversion into PI(4,5)P₂ exclusively in the outer leaflet mediated by PIP5K1C, ATP and Mg²⁺.

7.) *Effects of Mg²⁺ ions on the analysis of FGF2-GFP binding to PI(4,5)P₂ liposomes as analysed by FACS assays*

The experimental conditions mentioned by the Reviewer referring to Fig. 2B (FACS read-out) versus Figs. 1C and 3A (Biochemical sedimentation assays) are different. Only the FACS read-out is affected by Mg²⁺ ions whereas the sedimentation assay is not. This demonstrates that it is the way of quantification/data analysis of FGF2-GFP binding to PI(4,5)P₂ liposomes that causes this difference with the FACS assay being affected due to clustering of PI(4,5)P₂ liposomes, a phenomenon that affects data normalization. Nevertheless, both experimental approaches are useful and informative. However, as already outlined in

previous points of our response, using the FACS assay, absolute values for FGF2 binding efficiencies cannot be compared directly for conditions that either contain or do not contain Mg^{2+} ions.

8.) *Potential phase separation observed in GUVs shown in Fig. 4*

The Reviewer is right in pointing out this phenomenon that is visible in Fig. 4. We noticed this as well, however, so far did not discuss the matter in detail in the original manuscript because we did not observe FGF2-GFP binding to phase-separated membrane subdomains of GUVs on a consistent basis. For example, the GUVs shown in Fig. 3B and 3C show homogenous binding of FGF2-GFP to GUVs across the whole area of their membrane surfaces. Sometimes, we also observed FGF2-GFP phase separation to disappear once FGF2-dependent membrane pore formation had occurred. Nevertheless, we believe these observations could be indeed interesting as we have hypothesised the FGF2 membrane translocation machinery to be housed in cholesterol-rich, liquid-ordered nanodomains in previous studies (Lolicato and Nickel, 2022). Therefore, we take the current observations as an interesting starting point that we will investigate in detail in future studies. However, for the current study, these observations do not compromise our conclusions in any way. In the revised manuscript, we have discussed these observations in the context of a potential role of liquid-ordered nanodomains in FGF2 membrane translocation and the perspectives to address this hypothesis in future studies.

9.) *Analysis of FGF2-induced membrane pore formation in symmetric versus asymmetric PI(4,5)P₂ GUVs at a common time point of 40 min*

We agree with the Reviewer in that the data shown in Fig. 4 could be extended by a data set of a larger number of GUVs analyzed for membrane pore formation at fixed time points to foster the kinetic analysis that is shown so far in Figs. 4A and 4B along with quantification shown in Fig. 4C. Therefore, we did new experiments comparing symmetric versus asymmetric PI(4,5)P₂ GUVs, quantifying the percentage of GUVs characterized by FGF2-induced membrane pores as indicated by the luminal localization of the fluorescent tracer. As shown in the **new Fig. 5**, at 40 min of incubation, the percentage of GUVs with membrane pores was substantially higher for the population of asymmetric PI(4,5)P₂ GUVs when compared to those with a symmetric transbilayer distribution of PI(4,5)P₂. Along with a proper statistical analysis, the new data corroborate the central conclusion of our manuscript, demonstrating the asymmetric distribution of PI(4,5)P₂ to enhance the kinetics of FGF2 membrane translocation.

10.) *Lipid specificity controls in cell-based experiments quantifying FGF2 secretion under conditions compromising transbilayer asymmetry of PI(4,5)P₂*

We agree with Reviewer 1 that, in addition to PS used as a negative control, the use of other phosphoinositides more closely related to PI(4,5)P₂ would further strengthen our analysis. Therefore, in addition to PS, we did new experiments using PI(4)P, PI(3,4)P₂ and PI(3,4,5)P₃ added to cells followed by quantification of FGF2 membrane translocation to cell surfaces. As outlined in the revised manuscript, alike PS, adding PI(4)P to cells did not inhibit FGF2 secretion. However, similar to PI(4,5)P₂, the addition of PI(3,4)P₂ and PI(3,4,5)P₃ caused a substantial reduction in FGF2 membrane translocation. These new insights are quite interesting as we showed in previous studies that, albeit with lower efficiency, FGF2 does not only bind to PI(4,5)P₂ but also to PI(3,4)P₂ and PI(3,4,5)P₃ (Temmerman et al, 2008, Traffic). By contrast, as shown in both previous publications and in the current study, FGF2 does not bind to PI(4)P. As discussed in the revised manuscript, these new findings imply that the steep negative charge gradient produced by the asymmetric transbilayer distribution of PI(4,5)P₂ in native cells plays a key role in FGF2-induced membrane pore opening. In addition, it is also possible that phosphoinositides with multiple

negative charges such as PI(3,4,5)P₃ added to cell surfaces repel heparan sulfate chains from Glypican-1, an effect that may inhibit FGF2 membrane translocation in a severe manner. The data from these experiments including a proper statistical analysis are shown in the **new Fig. 9** and are discussed in detail in the revised manuscript.

Reviewer #2

Main points

- 1.) *Additional negative controls in FGF2-dependent membrane pore formation such as a catalytically inactive form of PIP5K1C or a kinase inhibitor*

This is a relevant question and we would like to emphasize that, in the absence of FGF2, the mixture of PIP5K1C, ATP and Mg²⁺ did not exert any pleiotropic effects of membrane integrity of GUVs either containing PI(4,5)P₂ or PI(4)P (Fig. 3D and 3E). In the revised manuscript, we added an additional control, the incubation of PI(4)P-containing GUVs with PIP5K1C and Mg²⁺ ions in the absence of ATP (**new Fig. 4D**). This condition mimicks a commercially available PIP5K1C inhibitor that targets the ATP binding site in a competitive manner. In this way we avoided the addition of another component such as a small molecule inhibitor or a mutant form of PIP5K1C with unknown properties towards the membrane integrity of GUVs. Omitting ATP in the continued presence of PIP5K1C and Mg²⁺ ions prevented the conversion of PI(4)P into PI(4,5)P₂, resulting in the absence of FGF2-dependent membrane pore formation (**new Fig. 4D**). We therefore consider our experimental system to be highly reliable and the data set of the in vitro experiments shown in Figs. 1 to 5 to be conclusive. Furthermore, the conclusions from the biochemical part of this study are corroborated by a strong data set from cell-based experiments, demonstrating transbilayer asymmetry of PI(4,5)P₂ to be a critical parameter for unconventional secretion of FGF2 from cells (Figs. 6 to 10).

- 2.) *Intensity changes in FGF2-GFP binding to LUVs containing PI(4)P and treated with PIP5K1C, ATP and Mg²⁺ in group d of Fig. 2B*

As also outlined in our response to Reviewer 1 [Point 6.); see above], the signals from the FACS read-out shown in grey bars (before reformation of LUVs) and light red bars (after reformation) are not directly comparable as Mg²⁺ ions (that affect the absolute FACS signals for FGF2-GFP binding) are removed during the liposome reformation procedure. The relevant comparison is therefore the FGF2-GFP binding signal for the light red bars in group d (asymmetric PI(4,5)P₂) versus light red bars in group a and c (symmetric PI(4,5)P₂). Here, for the light red bars, the signal drops substantially from group a/c to group d since the reformation procedure causes a dilution of PI(4,5)P₂ only for asymmetric PI(4,5)P₂ LUVs. To further analyse this phenomenon, we generated a standard curve of FGF2-GFP binding to LUVs as a function of the mol percentage of PI(4,5)P₂ (Fig. 2C). The signal obtained for reformed LUVs from group d (light red bars; Fig. 2B) lies between 2 and 3 mol% PI(4,5)P₂ of the standard curve shown in Fig. 2C. This is in line with a surface reduction of PI(4,5)P₂ from about 5 mol% before reformation to about 2.5 mol% after reformation. These data are consistent with the asymmetric nature of LUVs in which 5 mol% PI(4)P was added during formation that were subsequently converted by PIP5K1C, ATP and Mg²⁺ into PI(4,5)P₂ exclusively on the outer leaflet of GUVs.

3.) *Analysis of FGF2-induced membrane pore formation in symmetric versus asymmetric PI(4,5)P₂ GUVs at a common early time point*

A similar point was made by Reviewer #1 [Point 9.]; see above]. We agree with both Reviewers in that the data shown in Fig. 4 could be corroborated by a data set of a larger number of GUVs analyzed for membrane pore formation at a fixed time point to foster the kinetic analysis that is shown so far in Figs. 4A and 4B along with quantification shown in Fig. 4C. Therefore, we did new experiments comparing symmetric versus asymmetric PI(4,5)P₂ GUVs with regard to the percentage of GUVs characterized by FGF2-induced membrane pores as indicated by the luminal localization of the fluorescent tracer. As shown in the **new Fig. 5**, at 40 min of incubation, the percentage of GUVs with membrane pores was substantially higher for the population of asymmetric PI(4,5)P₂ GUVs when compared to those with a symmetric transbilayer distribution of PI(4,5)P₂. The new data corroborate the central conclusion of our manuscript, demonstrating the asymmetric distribution of PI(4,5)P₂ to enhance the kinetics of FGF2 membrane translocation.

Minor points

1.) *Physiological concentration of PI(4,5)P₂ in the inner plasma membrane leaflet of native cells*

Various studies have estimated the mol percentage of PI(4,5)P₂ in the inner plasma membrane leaflet of native cells to be around 1-5 mol% (reviewed in Wen et al 2021, Ann Rev Biochem). In in vitro experiments contained in publications from other laboratories and those from our group including the current study, PI(4,5)P₂ has been typically used at 2 to 5 mol% (e.g. Temmerman et al 2008, Traffic; Steringer et al 2012, J Biol Chem). We think using a PI(4,5)P₂ concentration in the upper range compared to what has been estimated in cells is justified since in vitro systems do lack some of the features and components of native membranes, typically requiring purified components to be included at slightly higher concentrations. While one of course aims at coming close to the conditions of native membranes, the reductionist approach allows to pinpoint the functions of individual components with the current study being a good example.

2.) *Use of LUVs versus GUVs in different experiments in this study*

As requested by the Reviewer, we have edited the revised manuscript explaining the used acronyms LUV (Large unilamellar vesicles) and GUV (Giant unilamellar vesicles). We used these different types of liposomes for different purposes. LUVs were mainly used to develop the experimental procedure to enzymatically convert PI(4)P in to PI(4,5)P₂, taking advantage of the possibility to form large amounts of such vesicles for biochemical experiments. The procedure was then transferred to GUVs that can be studied by confocal microscopy, a technique that we have used in a number of studies to analyze FGF2-dependent membrane pore formation at single GUV resolution. This approach allows for a kinetic analysis of this process which was key for the analysis of a potential role of PI(4,5)P₂ transbilayer asymmetry in FGF2 membrane translocation presented in this study. In the revised manuscript, we have improved all sections dealing with LUVs and GUVs, explaining the respective choices for different kinds of experiments.

3.) *Use of FGF2-GFP versus FGF2-Halo in biochemical in vitro experiments*

It is true that we used both FGF2-GFP and FGF2-Halo fusion proteins in various kinds of experiments. In principle, both kinds of fusion proteins work in all setups, demonstrating that our experimental systems do not depend on a particular kind of FGF2 fusion protein. Nevertheless, GFP and Halo differ in some properties with, for example, the Halo domain being strongly acidic (pI≈4.8). This causes FGF2-Halo to bind to

membranes a bit weaker than FGF2-GFP since the Halo domain is repelled from the negative headgroup charges of phospholipids (Lolicato et al 2023, J Cell Biol). At the same time, the acidic Halo domain reduces the tendency of liposomes to form clusters which is of advantage in biochemical in vitro studies involving liposomes. So basically, we optimized each of our experimental systems and used in each case the reagents that serve the purpose of the experiment best. Importantly, we did not encounter experimental observations that generally worked with only one or the other FGF2 fusion protein. In the revised manuscript, we have improved the introductions into the various experimental systems, explaining to the readers the purpose and the corresponding choices in terms of reagents in each section.

4.) *Statistical data in Figs. 1, 2 and 3*

A similar point has been made by Reviewer #1 [Point 3.]; see above], asking for a statistical analysis of the experiments shown in Figs. 1, 2 and 3. In the revised manuscript, we have **updated Figs. 1, 2 and 3** providing a statistical analysis including p-values to test whether differences observed for key conditions are statistically significant.

Reviewer #3

Major Comments

1.) *Mg²⁺ induced liposome clustering affecting absolute signals in the FACS-based read-out quantifying FGF2 binding to PI(4,5)P₂-containing liposomes*

A similar point has been raised by Reviewer #1 [Point 7.]) and Reviewer #2 [Point 2.]), see above. We would like to emphasize that the FACS-based assay to quantify protein-lipid interactions based upon lipid bilayers is a robust and reliable assay that we have originally established in the context of our discovery of FGF2 being a high affinity PI(4,5)P₂ binding protein (Temmerman et al, 2008). Since many experimental conditions such as the presence of bivalent cations can cause liposomes to cluster, the ability to detect such phenomena by the light scattering unit of the FACS setup is actually a strength of this assay. Simple approaches such as buffer exchange or using EDTA do not resolve this phenomenon as Mg²⁺ ions caused strong tethering of liposomes. The whole matter was tested intensively by dynamic light scattering (DLS) and visual inspection of Rhodamine-PE labeled liposomes. Furthermore, in a separated paper describing the method in detail, liposome clustering was detected by the light scattering unit of our FACS setup (Temmerman et al, 2009). Nevertheless, when using this assay in a proper way comparing data points only for identical experimental conditions, it accurately quantifies the relative amounts of FGF2 binding to PI(4,5)P₂-containing liposomes and, therefore, was used extensively in past studies (e.g. Lolicato et al 2023, J Cell Biol) and in the current study. Regarding the questions raised by all three Reviewers, we would like to emphasize that we interpreted the relative FGF2 binding signals only for experimental conditions that, from a technical point of view, were conducted under identical conditions and, therefore, could be compared directly. For example, those that were conducted in the presence of Mg²⁺ ions. Therefore, the data and the conclusions are accurate.

In addition, as an independent approach, in some of the experiments we conducted biochemical sedimentation assays, a semi-quantitative approach of measuring protein binding to lipid bilayers. This was done in Fig. 1 when we initially characterized the enzymatic conversion of PI(4)P into PI(4,5)P₂ and in Fig. 3 when we introduced the FGF2-Halo fusion protein. Beyond the FACS assay and the biochemical sedimentation experiments, we further analyzed FGF2-GFP binding to GUVs by confocal microscopy.

In summary, we feel that our experimental setup is well controlled with the conclusions being well supported by the presented data set. The revised manuscript has been improved with more detailed explanations on the properties of the three different assays and the choices we made in the various sections of this study.

2.) *Observed phase separation in GUVs shown in Fig. 4*

This phenomenon was also commented on by Reviewer #1 [Point 8.]; see above]. It is true that we observed apparent phase separation in the context of longer incubations when we recorded movies to determine the kinetics of FGF2-induced membrane pore formation. In some cases, phase separation disappeared after pore formation had occurred. By contrast, in experiments shown in Figs. 1 and 3, representing shorter incubations of FGF2-GFP with PI(4,5)P₂ containing GUVs, phase separation could not be observed. It is currently unclear whether the observed phenomenon is of relevance for the mechanism of FGF2-dependent membrane pore formation and membrane translocation. However, since we hypothesized the FGF2 membrane translocation machinery to reside in liquid-ordered nanodomains including the Na,K-ATPase, PI(4,5)P₂ and GPC-1 (Lolicato and Nickel, 2022), we will address this matter in future studies, analyzing the spatial organization of the FGF2 secretion machinery in terms of its lateral organization in plasma membranes. The revised manuscript has been edited in this regard, offering more background information on the implications the observations shown in Fig. 4 might have.

3.) *PI(4)P as an additional control in cell-based experiments shown in Figs 5 to 8*

As also outlined in our response to Reviewer #1 [Point 10.]; see above], we agree with both Reviewers that, in addition to PS used as a negative control, the use of other phosphoinositides more closely related to PI(4,5)P₂ would further strengthen our analysis. In the original manuscript, PS was chosen as an acidic phospholipid that is among the most abundant membrane lipids in the cytoplasmic leaflet of the plasma membrane. In the revised manuscript, we now present additional experiments with PI(4)P, PI(3,4)P₂ and PI(3,4,5)P₃ added to cells followed by quantification of FGF2 membrane translocation to cell surfaces (**new Fig. 9**). As outlined in the revised manuscript, alike PS, adding PI(4)P to cells did not inhibit FGF2 secretion. However, like PI(4,5)P₂, the addition of PI(3,4)P₂ and PI(3,4,5)P₃ caused a substantial reduction in FGF2 membrane translocation. These new insights are quite interesting as we showed in previous studies that, albeit with lower efficiency, FGF2 does not only bind to PI(4,5)P₂ but also to PI(3,4)P₂ and PI(3,4,5)P₃ (Temmerman et al, 2008, Traffic). By contrast, as shown in both previous publications and in the current study, FGF2 does not bind to PI(4)P. As discussed in the revised manuscript, these new findings imply that the steep negative charge gradient produced by the asymmetric transbilayer distribution of PI(4,5)P₂ in native cells place a key role in FGF2-induced membrane pore opening. In addition, it is also possible that phosphoinositides with multiple negative charges such as PI(3,4,5)P₃ added to cell surfaces repel heparan sulfate chains from Glypican-1, an effect that may inhibit FGF2 membrane translocation in a severe manner. The data from these experiments are shown in the **new Fig. 9** and are discussed in detail in the revised manuscript.

4.) *Quantification of fluorescent images shown in Fig. 6 (original manuscript numbering)*

We fully agree with the Reviewer 3. Quantifications have been performed and are now included in the revised version in the **new Fig. 7 (updated version of Fig. 6 from the original manuscript)**.

5.) *Statistics shown in Figs. 7 and 8*

We agree with Reviewer 3 and have adapted the statistical analysis of all cell-based experiments accordingly. The data are shown in the **new or updated figures 7, 8, 9 and 10**.

6.) *Blebbing phenomena observed in some recorded movies and their potential relevance for FGF2 membrane translocation*

We noticed those phenomena as well; however, they occurred on a rare basis and are not believed to play a role in FGF2 membrane translocation. The matter was investigated intensively in our initial studies introducing GUVs as a model system to analyze FGF2 membrane translocation in vitro (Steringer et al, 2012; Steringer et al, 2017). Also, in the context of cells, membrane blebbing has been shown not to be involved in unconventional secretion of FGF2 (Seelenmeyer et al, 2008). The revised manuscript has been edited, discussing the matter in the light of the relevant literature.

Minor Comments

1.) *Colour code missing in Fig. 2B*

We apologize for the inconvenience. At some point in the preparation of the first submission, the legend was lost unnoticed. The new version of this figure is now fully self-explanatory.

2.) *Concentration of doxycycline in cell-based experiments*

The Materials and Methods section has been adapted according to the request of the Reviewer, providing the concentration for doxycycline used in cell-based assays.

In conclusion, in the revised manuscript, we have addressed all points raised by the three Reviewers by new experimental data as well as textual adaptations and clarifications.

We would like to thank the Reviewers for their thorough analysis of our study, a detailed review that brought up important points that we hope to have addressed in a fully satisfying manner.

Many thanks again for considering our work. We hope you will find the revised version suitable for publication in *Nature Communications*.

Yours sincerely,